evolution, genetics, genomics

QTL mapping, genetic architecture, genetic correlation, heritability, ontogeny, *Pungitius*

**Author for correspondence:**
Antoine Fraimout
e-mail: fraimout.antoine@gmail.com

# Age-dependent genetic architecture across ontogeny of body size in sticklebacks

Antoine Fraimout[1], Zitong Li[1,2], Mikko J. Sillanpää[3] and Juha Merilä[1,4]

[1]Ecological Genetics Research Unit, Organismal and Evolutionary Biology Research Programme, Faculty of Biological and Environmental Sciences, University of Helsinki, FI-00014, Finland
[2]CSIRO Agriculture and Food, GPO Box 1600, Canberra, ACT 2601, Australia
[3]Research Unit of Mathematical Sciences, University of Oulu, FI-90014, Finland
[4]Area of Ecology and Biodiversity, School of Biological Sciences, The University of Hong Kong, Hong Kong SAR

AF, 0000-0003-4552-3553; JM, 0000-0001-9614-0072

Heritable variation in traits under natural selection is a prerequisite for evolutionary response. While it is recognized that trait heritability may vary spatially and temporally depending on which environmental conditions traits are expressed under, less is known about the possibility that genetic variance contributing to the expected selection response in a given trait may vary at different stages of ontogeny. Specifically, whether different loci underlie the expression of a trait throughout development and thus providing an additional source of variation for selection to act on in the wild, is unclear. Here we show that body size, an important life-history trait, is heritable throughout ontogeny in the nine-spined stickleback (*Pungitius pungitius*). Nevertheless, both analyses of quantitative trait loci and genetic correlations across ages show that different chromosomes/loci contribute to this heritability in different ontogenic time-points. This suggests that body size can respond to selection at different stages of ontogeny but that this response is determined by different loci at different points of development. Hence, our study provides important results regarding our understanding of the genetics of ontogeny and opens an interesting avenue of research for studying age-specific genetic architecture as a source of non-parallel evolution.

## 1. Introduction

How predictable is evolution? This question has intrigued evolutionary biologists for over a century [1–4], and although much progress has been made towards answering it, more remains to be learned about conditions and processes influencing the degree of predictability in evolutionary responses [5–8]. Understanding when and why evolution is predictable is not only of pure academic interest but also of practical utility. For instance, when dealing with the most pressing environmental problem of our times, global environmental change, there is a need to predict and understand if and how different species and populations will be able to adapt to changing environmental conditions [9].

Both empirical and theoretical work have established that when populations repeatedly and independently adapt to similar environmental conditions from the *same* pool of ancestral genetic variation, evolution is often (but not always) predictable: parallel adaptations evolve both at phenotypic and genetic levels [10]. A prime example of this is the loss lateral armour plates in the three-spined stickleback (*Gasterosteus aculeatus*) in response to colonization of freshwater habitats from marine environments; this adaptation is predictably underlined by parallel genetic changes in the Ectodysplasin (EDA) gene [11–13]. Inspired by this success story, the three-spined stickleback has become 'the' empirical model system to study predictability of evolution (e.g. [14–17]). However, when populations repeatedly and independently adapt to similar environmental conditions from *different* pools of ancestral genetic variation, evolution becomes much less predictable as

different sets of loci may underlie similar adaptive phenotypes in the different populations (e.g. [8,17]).

While a decade of evolutionary genetics research has explored factors influencing the probability of parallel evolution (e.g. [1,3,7,10,16,18–21]), and even demonstrated that phenotypic similarity can be underlined by different genes (e.g. [8,22–26]), less attention has been paid to age-dependent heterogeneity of genetic architecture as a potential source of nonparallel evolution at the genetic level.

Studies of both domesticated/laboratory [27–31] and wild populations [32–38] have shown that genetic variance of quantitative traits can change throughout development and that such changes may also be underlined by changes in the loci controlling trait expression [39,40]. This has potentially important implications for the study of parallel evolution as age-specific genetic architecture would constitute a heterogeneous pool of genetic variation for selection to act on. Because temporal variation in selection is common in the wild (reviewed in [41]) and that such selection can act on age-structured populations [42,43], a simple concept can be derived: if different genes control the expression of a quantitative trait at different ages and natural selection acts on the trait at different stages of ontogeny, the probability of parallel evolution would be lowered and by definition, so would the predictability of evolution.

The main objective of this study was to explore this idea by estimating genetic variance, heritability and the contributions of different chromosomes to body size variation over ontogeny in three nine-spined stickleback (*Pungitius pungitius*) crosses. By combining quantitative genetic and QTL-mapping approaches we tested the hypothesis that the genetic architecture of body size of *P. pungitius*—an important life-history trait—varies throughout ontogeny thus providing a heterogeneous source of genetic variation for selection to act on. We tested this hypothesis by evaluating the following assumptions: (i) Significant additive genetic variance and heritability of body size across ages would suggest that this trait can respond to directional selection at different stages of the ontogeny. (ii) Non-significant genetic correlations between sizes at different ages would imply heterogeneous genetic basis for this trait throughout development, and that selection acting at a specific age should not affect size at another age. (iii) If the same chromosomes contribute to body size variation at different ages and in different crosses, this would suggest that the evolutionary outcome of selection on body size would be predictable irrespectively of population- or age-specific selection. If on the other hand, different chromosomes underline variation in the body size at different ontogenetic stages and crosses, this would suggest that outcome of similar selection pressure in respect to ontogeny and population would be unpredictable at genetic level.

## 2. Material and methods

### (a) Study species, phenotype and genotype data
The nine-spined stickleback is a teleost fish with a wide distribution range across the Northern Hemisphere [44]. In Europe, isolated pond populations of *P. pungitius* [45–47] have repeatedly evolved extreme morphological [8,48] and behavioural [49] phenotypes, making them particularly interesting for the study of parallel adaptive evolution [8,17,50]. Marine and freshwater *P. pungitius* display also contrasting growth trajectories: while

the former ecotype shows fast growth to an early maturation at small size, the latter exhibits prolonged growth and delayed maturation at larger size [51].

Here, we analysed growth trajectories in three $F_2$ marine-freshwater crosses used in previous studies exploring the genetic architecture of various quantitative traits [8,50–52]: HEL × RYT, HEL × PYÖ and HEL × BYN. Briefly (but see electronic supplementary material and [53]), grandparental ($F_0$) individuals were collected from the wild and mated in the laboratory to produce $F_1$-offspring. $F_2$-offspring were produced by mating single randomly chosen pairs of $F_1$ in each cross (see electronic supplementary material) and in total, 274 $F_2$ offspring were obtained for the HEL × RYT cross, 278 for the HEL × PYÖ cross and 307 for the HEL × BYN cross.

Growth data for all individuals was obtained by measuring the body size of each fish at different time points throughout development. Individuals from HEL × PYÖ and HEL × BYN crosses were measured at nine time points at 4, 8, 12, 16, 20, 24, 28, 32 and 34 weeks post-hatching. Fishes from HEL × RYT were measured at seven different time points 2, 6, 10, 14, 18, 22 and 26 weeks post-hatching. Body size was measured as the distance between the tip of the snout and the base of the posterior end of the hypural plate for all individuals from digital photographs including a millimetre scale and using the tps.Dig software [54]. All individuals were sequenced using the restriction-site associated DNA approach (RADseq; [55]) to obtain a panel of single nucleotide polymorphisms (SNPs) as described in [56] (see electronic supplementary material). The final genomic dataset consisted of 21 832 SNPs for HEL × RYT; 21 747 SNPs for HEL × PYÖ and 21 747 SNPs for HEL × BYN.

### (b) Phenotypic variation in growth trajectories
We first investigated patterns of growth at the phenotypic level within each cross to (i) determine whether individuals had reached their adult size, and (ii) to describe the growth trajectories explaining phenotypic variation in our datasets. To this end, we first applied a von Bertalanffy growth curve model [57] as

$$y_t = y_{inf}(1 - e^{-k(t-t_0)}), \tag{2.1}$$

where $y_t$ is the length of an individual at age $t$, $k$ the intrinsic growth rate, $t_0$ is the estimated hypothetical length at age $t = 0$, and $y_{inf}$ is the asymptotic length, corresponding to the estimated final body size for each individual. Second, we applied an Infinite Dimensional Model (IDM; [58,59]) using the *InfDim* R package [59]. This approach uses the phenotypic covariance matrix of age-specific body sizes to determine the main growth trajectories underlying the phenotypic variation in the data (see electronic supplementary methods for details). In other words, the IDM allows to describe at the phenotypic level whether growth is uniform among individuals (i.e. all individuals increase proportionally in size with age) or if alternative growth trajectories are found among individuals (e.g. large individuals at young age are small at old age). We used the *IDM.bootCI* function on each cross separately to obtain the 95% confidence intervals (CIs) around the estimated parameters via bootstrapping.

### (c) Quantitative genetic analyses: genetic variance and heritability
Analytically, longitudinal data such as growth measurements can be viewed either as a character taking on a different value at each discrete age (character state trait), or time-dependent observations describing a continuously varying trajectory ('function-valued trait'; [58]). Here we used both approaches to analyse

the data under a Bayesian framework and estimated the quantitative genetic parameters of body size throughout growth.

First, we partitioned the phenotypic variance ($V_P$) of age-specific body size into its additive genetic component ($V_A$) by fitting an animal model of the form

$$y = X\beta + Z\gamma + \varepsilon, \tag{2.2}$$

where $y$ is the vector of phenotypic values for age-specific body size, $\beta$ is the vector of fixed effect, $\gamma$ is the vector of random effects, $\varepsilon$ is the vector of residual errors and $X$ and $Z$ are the design matrices relating to the fixed and random effects, and corresponding to the individual values of body size at one age and the matrix of relatedness coefficients, respectively. The latter describes the genetic relatedness among sampled individuals and can be constructed from pedigree relationships (i.e. theoretical relatedness coefficients) or by estimating the proportion of genome shared identically-by-descent among individuals from SNP markers (i.e. realized relatedness coefficients) [53]. Here, we used the Genomic Relationship Matrix (GRM) constructed from SNP markers (see electronic supplementary material, methods) in order to improve the accuracy of variance component estimation [53].

Each animal model was fitted using the *MCMCglmm* R package [60] using flat priors (setting the degree of belief parameter *nu* to 0) for 1 030 000 Monte Carlo Markov chain (MCMC) iterations with a burn-in period of 30 000, thinning every 1000th iteration and by adding sex of the individuals as a fixed effect. Following the character state approach, body size was modelled as a Gaussian response variable using the *family* option of the *MCMCglmm* function and we fitted separate univariate animal models for size at each age in all crosses. We calculated heritability of age-specific body size from the posterior distribution of the 1000 MCMC samples as

$$h^2 = \frac{V_A}{V_p}, \tag{2.3}$$

and computed the respective 95% highest posterior density (HPD) intervals using the *HPDinterval* function in *MCMCglmm*. To allow for meaningful comparison of the magnitude of additive genetic variance across ages, we calculated the coefficients of additive genetic variation ($CV_A$) [61] as

$$CV_A = 100 \times \frac{\sqrt{\sigma^2_{BSA}}}{\overline{BSA}}, \tag{2.4}$$

where $\sigma^2_{BSA}$ is the additive genetic variance for body size at age (BSA) and the denominator $\overline{BSA}$ is the phenotypic mean of the body size at a specific age. To compare these results to the 'function-valued trait' approach (see below), we applied a penalized spline smoothing function in R to the full set of MCMC posterior (the 'character-state' estimations) to infer the patterns of changes in $V_A$ and $h^2$ with age occurring among and between each time point.

We then re-estimated $V_A$ and $h^2$ following the 'function-valued trait' approach by using the method described in [62] implemented in the *dynGP* package (https://github.com/aarjas/dynBGP).

This method differs from the 'character-state' approach by considering the full distribution of trait values across all ages rather than estimating parameters at each age separately. As such, the method allows to model dynamic variance components and $h^2$ for longitudinal data. Similarly, to the animal models described above, the *dynGP* analysis uses the GRM constructed from SNP data to account for relatedness among individuals and estimate $V_A$ and $h^2$ under a linear mixed model framework [62]. We followed the methodology of Arjas *et al.* [62] and ran separate models for each cross for 1 000 000 MCMC-iterations (see electronic supplementary material, methods for details).

As the model formulation in *dynGP* does not allow for fixed effects we used vectors of residuals from linear regressions of age-specific body size on sex as response variables in order to have equivalent structures to the animal models described above.

## (d) Quantitative genetic analyses: genetic correlations and **G** matrix

To estimate genetic correlations across different ontogenetic time-points, we analysed body size across all ages by fitting a multivariate version of the animal model described in equation (2.2). Because our growth data correspond to repeated measurements of the same individuals over time, we added a permanent environment effect term in the animal model as a random effect to represent the dependent part of the residuals (caused by repeated measurements). From the multivariate models, genetic correlations among body size across ontogeny were calculated as

$$r_G = \frac{Cov_{Aij}}{\sqrt{V_{Ai} \times V_{Aj}}}, \tag{2.5}$$

where $Cov_{Aij}$ is the additive genetic covariance between traits $i$ and $j$ and $V_{Ai}$ and $V_{Aj}$ the additive genetic variances for traits $i$ and $j$, respectively [63]. These multivariate animal models further allowed us to estimate the genetic (co)variance matrix (**G**) of body size at different ages in each cross. The elements of **G** capture the additive genetic variance (diagonal) and covariance (off-diagonal) underlying the expression of multivariate quantitative traits [64]. In our case, **G** of age-specific body size summarizes the additive genetic variance of body size at each age (similarly to our univariate animal models) as well as the covariance between them. Eigen-analysis of **G** allows identification of the major axes of genetic variation in the data (the principal components [PC] or eigenvectors of **G**) and the correlation between the traits and these axes of variation (the trait loadings). Thus, we performed a principal component analysis (PCA) on each **G** estimated from the multivariate animal models for each cross separately. This allowed us to (i) estimate the percentage of variance explained by the first PC of each **G** (PC1 or $\mathbf{g}_{max}$) and thus, estimate the amount of additive genetic variance underlying growth, and (ii) calculate the loadings of age-specific body size on $\mathbf{g}_{max}$. This in turn informs us of the correlational structure of body size throughout ontogeny: positive loadings of body size at each age on $\mathbf{g}_{max}$ would be indicative of positive genetic correlations between age-specific sizes (integration of body size ontogeny; [33]); conversely, a change in signs of the loadings along $\mathbf{g}_{max}$ would be indicative of a modular ontogeny and a loss of genetic correlations between age-specific body sizes [33].

Due to the computational burden of MCMC sampling in multivariate animal models using GRM, we reduced the number of observations in HEL × PYÖ and HEL × BYN from nine to seven by removing measurements from age at Week 24 and Week 32. For the same reason, models were run for 203 000 MCMC iterations with a burn-in period of 3000 and thinning every 100th iteration. To confirm the results from univariate models, $V_A$ and $h^2$ of age-specific body sizes were also computed directly from the multivariate models.

## (e) QTL mapping

We used a single-locus mapping approach [65–67] to identify the genomic regions underlying variation in body size at each ontogenetic time point (see detailed models in electronic supplementary material). In order to separate the dam and sire alleles—and to obtain additional information of the origin of the QTL effects—we incorporated parental phasing in our QTL

model. To this end, we used the data produced in [8] for the same three F₂ crosses. Briefly, parental and gran-parental phase were obtained from dense SNP data (see detailed description in [8,65]) using LEP-MAP3 [68], and a linkage disequilibrium (LD) network-based dimensionality reduction (LDna; [69]) was applied to decrease redundancy in the data due to linkage. For each LD-cluster comprising sets of highly correlated SNPs genetic information was extracted using PCA and the PC-coordinates from the first axes explaining the largest proportion of variation were used for QTL mapping. For each cross, we applied our QTL model separately to the phenotypic vectors of size at each age and using the complexity reduced SNP panel after correcting for the effect of sex and subsequently estimated the proportion of variance explained (PVE) by all QTL (see electronic supplementary material). We also applied the mapping model to the two parameters $k$ and $L_{inf}$ estimated from the Von Bertalanffy growth curve model described in equation (2.1). This allowed us to test whether or not there is any global QTL associated with growth in multiple time points. For each QTL model, a permutation procedure (10 000 permutations) was used to account for multiple testing [65,70].

## 3. Results

### (a) Phenotypic variation in growth trajectories

The asymptotic sizes estimated from the Von Bertalanffy growth curves were similar to the mean body size at the oldest age in HEL × BYN and HEL × PYÖ crosses (electronic supplementary material, figure S1 and table S1), suggesting that the data accounted for most of the growth in the measured individuals. By contrast, asymptotic size was higher than mean size at the end of the experiment in HEL × RYT ($L_{inf} = 55.657$ [54.889–56.509]; $L_{max} = 48.589$ [48.088–49.090]; electronic supplementary material, table S1). Results from the IDM indicated that variation in ontogeny of body size in all crosses was mainly explained by a growth trajectory (i.e. an eigenvector of the phenotypic variance–covariance matrix) predicting an increase in size with age and accounting for 65–80% of the total variation (electronic supplementary material, figure S1). In all crosses, a second growth trajectory described a negative correlation between early and late growth (electronic supplementary material, figure S1) indicating that some individuals that were larger at an early age tended to be smaller later in life, and *vice versa*.

### (b) Quantitative genetic analyses: genetic variance and heritability

In all crosses and at all ages through ontogeny, there was additive genetic variance and heritability in body size (figure 1; electronic supplementary material, figures S2–S5). Additive genetic variance increased with age in all crosses and $V_A$ for body size was significantly higher at the last ontogenetic time-point compared to the first in HEL × PYÖ and HEL × RYT crosses (electronic supplementary material, figures S2–S5 and table S2). Phenotypic and residual variance also increased with age in all crosses (electronic supplementary material, figures S2–S5 and table S2), and consequently, heritability of body size remained relatively constant throughout ontogeny in all crosses and did not significantly differ between ages (figure 1; electronic supplementary material, figures S2–S5; table S2).

### (c) Quantitative genetic analyses: genetic correlations and **G** matrix

Genetic correlations between sizes at consecutive ages were positive in all crosses: size at age $n$ was positively correlated to size at age $n + 1$ and $n - 1$ (figure 1). Genetic correlations decreased with increasing distance between time-points in all crosses (figure 1). In HEL × PYÖ, genetic correlations were no longer statistically different from zero between the size at Week 4 and size at Week 20 and onwards (figure 1). In HEL × BYN, body size at first time-point was not genetically correlated to body size at the following time-points (figure 1). In HEL × RYT genetic correlations decreased with age and body size at the first time-point was not correlated with body size at Week 14, and weakly correlated with body size at Week 18 to 26 (figure 1).

In all crosses, the first principal component of **G** explained most of the genetic variance in growth (table 1), further confirming ample additive genetic variation for body size throughout ontogeny in all crosses. In HEL × PYÖ, a sign change in the loading coefficients of $\mathbf{g}_{max}$ indicate that early (Week 4, Week 8) and later (Week 16 onwards) sizes are not genetically correlated and that ontogeny is modular between early and late growth. In HEL × BYN, the same result was observed with a sign change of the loading coefficients of $\mathbf{g}_{max}$ at Week 20. In HEL × RYT loadings were positive for all ages-specific sizes, indicating positive genetic covariation between sizes throughout ontogeny. In HEL × PYÖ and HEL × BYN, a substantial amount of variance (9.73% and 12.36% respectively; table 1) was also explained by the PC2 where a sign change of the loading coefficients was also observed.

### (d) QTL-mapping

A total of 14 unique QTL associated with body size variation at different ontogenetic stages were detected (figure 2; electronic supplementary material, table S3 and figures S6–S8). We did not find any global QTL underlying the growth parameters $k$ and $L_{inf}$ in the HEL × BYN cross (electronic supplementary material, figure S9). A single QTL was detected in HEL × PYÖ for $k$ on Chr. 1 (electronic supplementary material, figure S9) and similarly for HEL × RYT, a single QTL controlling for asymptotic size ($L_{inf}$) was found on Chr. 7 (electronic supplementary material, figure S9). This shows that different QTL underline expression of body size not only among the different crosses but also between different ontogenetic stages within the crosses (figure 2; electronic supplementary material, table S2). Most of the QTL effects traced down to the sire (M, in figure 2; electronic supplementary material, table S2) indicating that the observed allelic effects were segregating in the male F₀ grandfather (pond individual). One peak indicative of segregation in the F₀ female (marine individual) on Chr. 18 (at age Week 24 in.HEL × BYN), Chr. 9 and Chr. 13 (at age Week 12 in.HEL × PYÖ; figure 3b), and another on Chr. 13 (at ages Week 10 and 14 in.the HEL × cross; figure 2c) were found in addition to the QTL for $k$ and $L_{inf}$ in HEL × PYÖ and HEL × RYT, respectively. Dominance effects were rare: only a single significant dominance allelic effect on Chr. 1 at age Week 12 in.the HEL × PYÖ cross was observed (figure 2b).

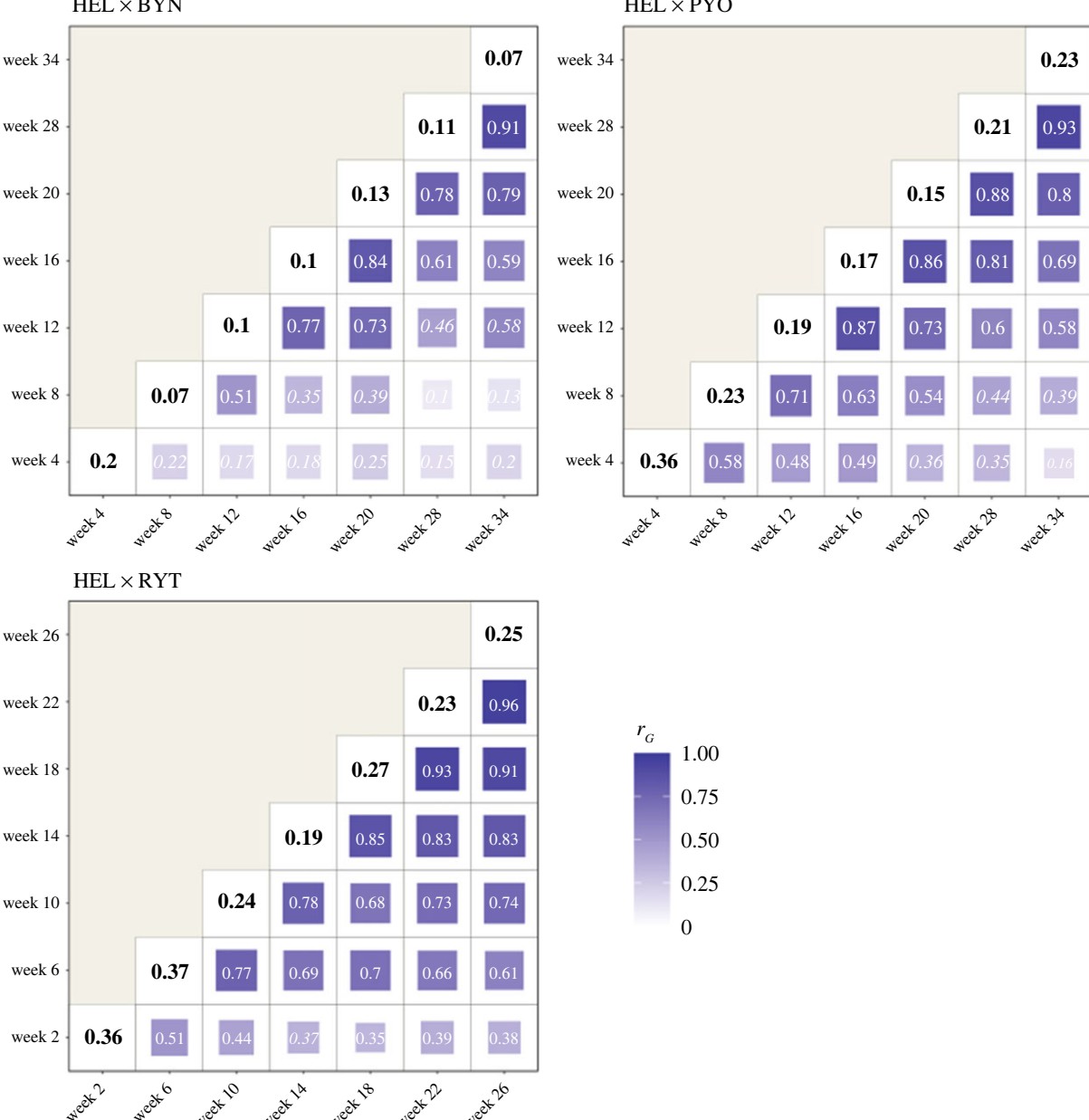

**Figure 1.** Heritability and genetic correlations across ages. Strength of the genetic correlations ($r_G$) of body size across ages is represented by the size and colour of each tile of the heatmap. Significance of the genetic correlation estimates is indicated in bold white text and non-significant values in italic white text. The posterior mode of the heritability for body size estimated from the multi-trait model (see 'Material and methods' and electronic supplementary material, figure S3) at each age is shown in the diagonal for each cross (black text). (Online version in colour.)

## 4. Discussion

Heterogeneous genetic underpinnings for similar phenotypic adaptations are evidence of redundancy of organismal genetic architectures: similar phenotypic end-points can be reached with diverse genetic mechanisms. Our results demonstrate that the QTL contributing to size were different across ages both within and among the three crosses studied here. These results were supported by analyses of genetic correlations across the age groups; early and late growth genetic correlations were generally weak or not significantly different from zero, and always significantly lower than one. These results align with earlier findings from this study system showing that heterogeneous genetic architectures underlie similar phenotypic adaptations in different pond populations [8]. The main novelty in the current results is that they demonstrate how ontogenetic heterogeneity in genetic

architecture of an important life-history trait may influence local adaptation: were selection to act on size at different points of development, genes governing the expected selection responses would be very different. In the following, we will discuss these and related points in more detail.

Within each of the three crosses, different QTL were often associated with body size at different ages, and the significant QTL effects were predominantly associated with alleles derived from the pond grandparent, and seldom with alleles from the marine grandparent. The latter finding makes sense in the light of the fact that pond fish have been shown to be consistently larger than marine fish throughout development [52], and therefore, one would expect pond alleles to contribute to size more strongly than the marine alleles. This could also indicate that most of the additive genetic variation observed results from the segregation of pond alleles. In a few cases, QTL effects tracing to both parents were observed,

**Table 1.** Percentage of variance explained and trait loadings on the two first eigenvectors of the principal components analysis of G. The percentage of genetic variance in growth trajectories explained by the first two eigenvectors of the principal components analysis of the G matrix is shown for each cross. Trait loadings on $g_{max}$ are shown for each age-specific body size.

| | HEL × BYN | | HEL × PYÖ | | HEL × RYT | |
|---|---|---|---|---|---|---|
| | PC1 ($g_{max}$) | PC2 | PC1 ($g_{max}$) | PC2 | PC1 ($g_{max}$) | PC2 |
| % var. explained | 81.76% | 12.36% | 87.37% | 9.73% | 95.48% | 2.72% |
| age 1 | −0.097 | 0.200 | −0.065 | 0.050 | 0.015 | −0.015 |
| age 2 | −0.191 | −0.340 | −0.036 | 0.471 | 0.134 | −0.659 |
| age 3 | 0.074 | −0.627 | 0.137 | 0.621 | 0.185 | −0.680 |
| age 4 | 0.260 | −0.489 | 0.263 | 0.480 | 0.295 | −0.153 |
| age 5 | 0.406 | −0.321 | 0.415 | 0.198 | 0.450 | 0.153 |
| age 6 | 0.570 | 0.113 | 0.556 | −0.099 | 0.547 | 0.168 |
| age 7 | 0.626 | 0.309 | 0.652 | −0.334 | 0.599 | 0.166 |

and only in one case was there evidence for dominance contribution. Hence, these findings suggest that the contribution of dominance effects is small and that most observed allelic effects were additive, and mostly coming from pond genetic background.

For polygenic traits, inherent pitfalls of QTL mapping analyses include low statistical power to detect significant QTL or conversely, detection of false positives [71]. Overall, the effect sizes and amount of PVE by the age-specific QTL within each cross were relatively low, suggesting that body size remains a largely polygenic quantitative trait in *P. pungitius*. Thus, the detection power of our QTL mapping analysis could very well be limited by sample size and statistical power to detect a sufficient number of causative loci at each age. Although we cannot refute this possibility with the data at hand, several results support an age-specific genetic architecture of body size in this species.

First, we did not find evidence for a permanent QTL affecting size at all ages within each of the three studied crosses and this result was also evidenced by the absence of significant QTL for the growth rate parameter *k*. Although a QTL on Chromosome 9 seemed to underline body size variation in consecutive ages in the HEL × BYN and HEL × PYÖ crosses, the allelic effects of these QTL were found to go in the opposite directions (as reflected by a sign change of effect sizes) suggesting this QTL is effectively different between the two populations. Second, genetic correlations between age-specific sizes tended to decay with time, and the correlations between early and late sizes were weak or non-existent. This was also manifested in the analyses of integration of the first eigenvector of **G**, which revealed evidence for modular genetic architecture of growth in two of the three crosses. This means that selection on size at any given age would not necessarily affect size across the whole ontogeny [58]. On the contrary, the results suggest that genes responsible for early and late growth are acting independently, and therefore, selection acting at early and later ages would be acting on different sets of genes. This result is also in line with previous findings from livestock studies showing that growth-related traits can have different genetic underpinnings at early and late ages [39,72–74]. For instance, a QTL mapping study using a F₂-intercross between

phenotypically distinct chicken (*Gallus gallus*) lineages revealed that largely different loci contributed to variance at the start and endpoints of the growth curve [75].

Finally, our relatively large family sizes allowed us to detect significant QTL explaining small amounts of genetic variance (i.e. PVE less than 0.1), suggesting that the study had reasonable statistical power. Furthermore, our LDn-based QTL-mapping analyses are statistically powerful in detecting QTL while avoiding false positives [65]. Hence, despite the relatively small effect sizes of the detected QTL there are strong ground to believe that the results reflect heterogeneity in genetic architecture among ages and crosses rather than statistical noise.

We also observed among cross heterogeneity in the age-specific QTL effects, suggesting that genetic architecture of body size variability might differ among the pond populations similarly as shown earlier for pelvis reduction in these populations [8]. However, some caution is warranted in interpretation of the observed population differences in QTL locations for at least two reasons. First, while exact time-points of measurement were the same two of the three crosses, fish in one of the crosses were measured in slightly different time-points than in the two others. Given that we picked up time-dependent variation in QTL effects, this could have influenced which QTL were detected in one of the crosses. However, the similarly measured crosses showed different patterns of QTL expression. Second, it should be kept in mind that all three crosses were established from a single pond and a single marine (grand) parent, and the crosses are therefore unlikely to carry all relevant allelic variation present in the populations of origin. Hence, there might be a large stochastic component to QTL detectable in each of the three crosses. While we do not have data to refute this possibility, an earlier study using empirical and simulated data gave strong evidence for heterogeneous genetic architecture underlying pelvic reduction in these populations [8], thus suggesting that this stochasticity might not be of great concern. Irrespective of what explains heterogeneous QTL effects among different crosses, sampling of allelic variation in parental populations is irrelevant to interpretation of the heterogeneous QTL effects across ontogeny.

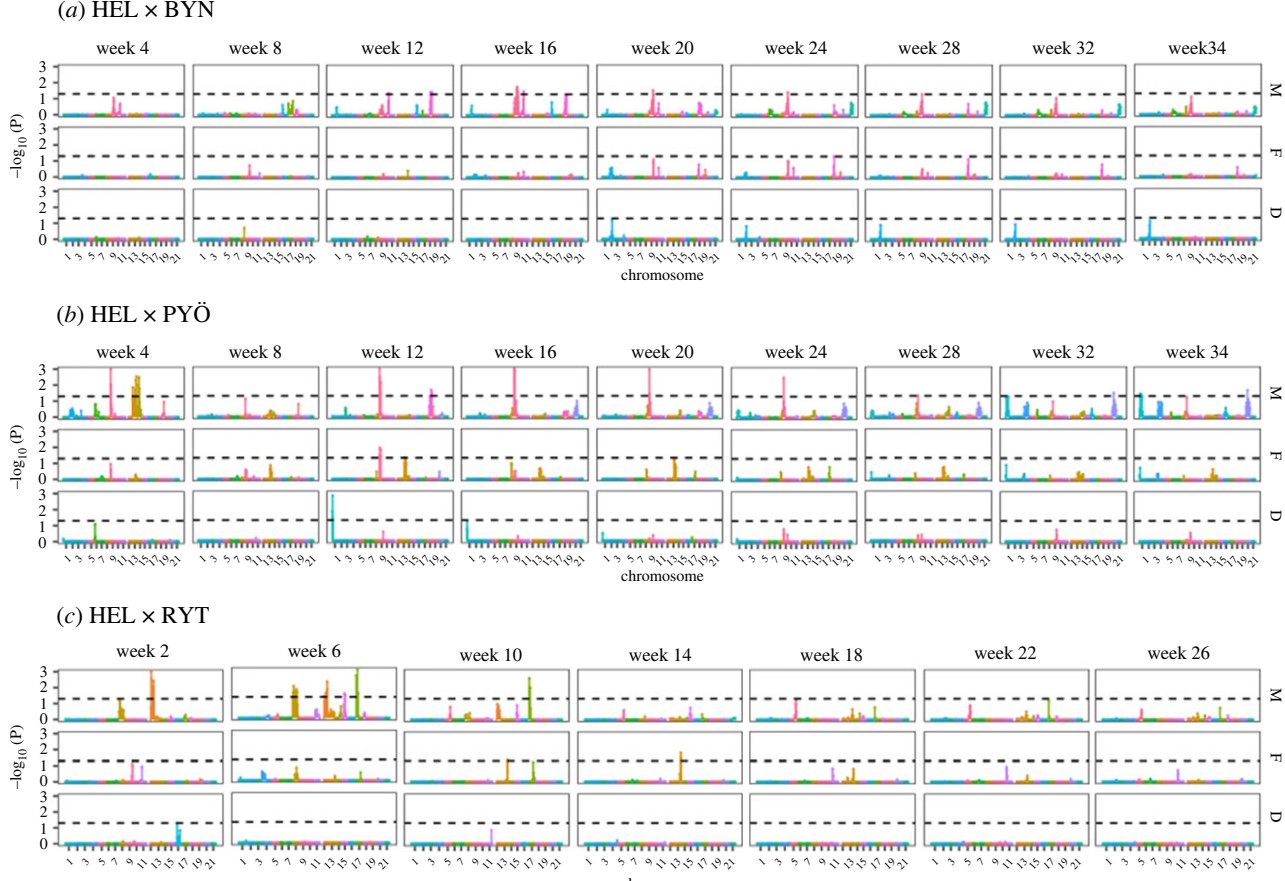

**Figure 2.** QTL-mapping results for age-specific body size. Results from the four-way QTL-mapping are shown for each cross (a–c) and for body size at each age (Week n). For each cross and each age, panels show whether the QTL are inherited from the sire (M) or the dam (F) along with the dominance effect (D) estimated from model [8] in the main text. Results are based on permutation and the significance threshold (dashed horizontal line) is shown on the logarithm scale ($-\log_{10}(P)$; $p = 0.05$). Colours represent different chromosomes. (Online version in colour.)

We found additive genetic variance and heritability for body size across ages in all three crosses. Our results suggest an increase in additive genetic variance with age as previously reported in several quantitative genetics studies of ageing (e.g. [34,76–80]). However, the confidence intervals around our estimates were large, precluding strong inferences on changes in genetic variance with age but nevertheless showing that body size is heritable throughout ontogeny in *P. pungitius*. Thus, regardless of the patterns of change with age, the heritable variation at each age in each cross suggests that body size can respond to directional selection at different stages of the ontogeny.

The variance component and heritability estimates in this study were based on single full-sib family per cross using variance in relatedness among sibs to estimate the causal components of variance. This approach has been used in earlier studies (e.g. [53,81]) and has the advantage that the estimates are not subjected to additional variance attributable to uncontrolled environmental effects [81,82]. Although other sources of non-additive variance such as dominance and maternal effects could influence our estimates [83], such effects are expected to be minimal for body size in both freshwater and pond populations of *P. pungitius* ([53,84], respectively). That said, since the analysed crosses are artificial $F_2$ generation inter-population crosses, the estimated quantitative genetic parameters may not be representative of those in wild populations.

In conclusion, our results show that age is a potential source of genetic heterogeneity for an ecologically important trait in a wild species. Although such heterogeneity in the genetic bases of quantitative traits had been observed in domestic species and brought forward as a way to manipulate and increase growth of livestock, its importance for local adaptation in the wild was largely omitted. As such, age-specific genetic architecture would decrease the predictability of evolution by increasing the number of genomic regions available for selection to act on and, consequently, decreasing the probability of parallel evolution. In sticklebacks, population-specific age structure and age-specific selection is expected to occur in the wild [85–87] and more generally, temporal variation in selection at different ages should be pervasive in nature [41]. We thus propose that the age structure of natural populations constitutes a promising parameter to account for in studies of parallel evolution, and that future research on the pervasiveness of age-specific genetic architecture in the wild should prove particularly useful in our ability to predict evolution.

Ethics. All experimental protocols were approved by permission (ESLHSTSTH223A) from the National Animal Experiment Board, Finland.

Data accessibility. Raw sequence reads have been submitted to NCBIs short-read archives with accession nos PRJNA673430 and PRJNA672863. Phenotype data are included as electronic supplementary material [88].

Authors' contributions. A.F.: conceptualization, data curation, formal analysis, validation, visualization, writing—original draft, writing—review and editing; Z.L.: data curation, resources, validation, writing—review and editing; M.J.S.: resources, validation, writing—review

and editing; J.M.: conceptualization, funding acquisition, investigation, project administration, supervision, writing—original draft, writing—review and editing.

All authors gave final approval for publication and agreed to be held accountable for the work performed therein.

Conflict of interest declaration. We declare we have no competing interests.

Funding. This study was supported by the Academy of Finland (grant nos. 129662, 134728 and 218343 to J.M.).

Acknowledgements. Thanks are due to Gabor Herczeg, Abigel Gonda, Yukinori Shimada, Mirva Turtiainen, Chris Eberlein, Takahito Shikano, Laura Hänninen, Kirsi Kähkönen, Miinastiina Issakainen and Sami Karja for fish breeding and DNA extractions, and Jing Yang for help in phenotyping the fish. The authors thank Petri Kemppainen and Emma Vatka for useful discussions on previous versions of the manuscript. The computing resource support from CSC, the Finnish IT Center for Science Ltd administered by the Ministry of Education and Culture, Finland, is gratefully acknowledged.

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
