## [Peer Review File · Proceedings of the Royal Society B: Biological Sciences]

Review History

RSPB-2021-0954.R0 (Original submission)

Review form: Reviewer 1

Recommendation

Accept with minor revision (please list in comments)

Scientific importance: Is the manuscript an original and important contribution to its field?

Good

General interest: Is the paper of sufficient general interest?

Good

Quality of the paper: Is the overall quality of the paper suitable?

Excellent

Is the length of the paper justified?

Yes

Should the paper be seen by a specialist statistical reviewer?

Yes

Do you have any concerns about statistical analyses in this paper? If so, please specify them explicitly in your report.

No

It is a condition of publication that authors make their supporting data, code and materials available - either as supplementary material or hosted in an external repository. Please rate, if applicable, the supporting data on the following criteria.

Is it accessible?

Yes

Is it clear?

Yes

Is it adequate?

Yes

Do you have any ethical concerns with this paper?

No

Comments to the Author

This paper does a careful study of the genetic basis of body size across ontogeny in a species of stickleback fish. The paper is motivated by the interest in predicting evolution in the sense that convergent evolution of phenotype or genetic architecture results from different conditions. The authors point out that even though predictability has been found to obtain when populations have the same genetic background, predictability may not obtain when the backgrounds differ. To assess predictability in an ontogenetic context, the authors estimate the additive genetic covariation of body size across ontogeny for three sets of sticklebacks, each set having a different genetic background (the grandfather comes from a different freshwater pond in each set). Additionally, the authors assess the genetic architecture of body size across ontogeny in the three sets of sticklebacks (QTL analysis). The authors find that additive genetic variation for body size across ontogeny is substantial, that heritability of body size remains roughly unchanged across ontogeny, and that loci contributing to body size vary substantially across ontogeny and among genetic backgrounds, despite rather subtle differences in body size among genetic backgrounds.

I find the paper is rigorous in its methods and interpretations and provides interesting data regarding heterogenous genetic architecture of body size over ontogeny depending on genetic background. I don't have any major concerns with the paper but make a few suggestions to help readers further.

The introduction is framed in terms of the predictability of evolution, but the discussion does not clearly say how the results relate to that original motivation. For instance, the conclusion revolves around how similar phenotypes may arise from different genetic architectures, and that selection on phenotypes stemming from different genetic architectures would lead to different genetic architectures (L 395-397), the latter of which sounds like a truism. Can you conclude more directly on predictability? If not, can you say why not?

Specific comments:

Title: the word "similar" is not clear in the title (similar to what?). It would be clearer to replace it by something like "ontogenetically invariant" or something like that. Also, adding something like "Variable" as the first word in the title would clarify the point. This would yield: "Variable age-dependent genetic architecture underlines ontogenetically invariant heritability of body size in sticklebacks".

L 24: This statement made me wonder if the authors are aware of this paper: Cheverud J. M. et al. 1983. Quantitative genetics of development: genetic correlations among age-specific trait values and the evolution of ontogeny. *Evolution*. 37, 895-905. The authors do not discuss or cite it, although that paper does very similar things to what this paper does.

L 27: I would rather say “roughly constant” or something like that rather than “constant” given the fluctuations in the mean and the wide confidence intervals.

L 54: “global change” would be better stated as “climate change” or “global climate change”.

L 68,69: I don't think the authors mean “positive function” here (any probability is a positive function whenever it is non-zero). Do you mean “increasing function”?

L 67-69: If “positive function” means “increasing function”, then this statement seems to say that gene flow (i.e., different pools) increases effective population size and so predictability. This seems to contradict the immediately previous statement in L 64-66 that evolution is less predictable with variation from different pools.

L75: I think it should be “attention has been paid to” rather than “on”.

L 96: I think the statement that the evolutionary outcome would be unpredictable is easily misunderstood if taken out of context. Here predictability has a narrow meaning (i.e., obtaining the same outcome from different conditions), but predictability in common speech has a broader meaning. So, stating that the evolutionary outcome is unpredictable is easily understood in the broader meaning (i.e., that you can't predict the outcome, but in principle you could if you had a lot more information about the system). Anyway, this is just to say that it would be good to add a parenthesis here reminding the reader that predictability here has a narrow sense (i.e., repeatability).

L 153: t_0 is not the estimated length at age 0, but “a value used to calculate size when age is zero” (phrasing taken from Wikipedia, Von Bertalanffy function article).

L 169: What are the X and Y measuring? Age and gene content, respectively? Some more information on what these matrices contain is needed here.

L175: “additive coefficients” of what? Are these regression coefficients of phenotype on SNP's, or something like that? Again, some more specification of W is needed.

L 183: Rather than “each size”, do you mean just “size”?

L 197: I think the statement “Contrary to the univariate animal model approach” is not helpful here. A univariate animal model would not even allow for the calculation of genetic covariances (there is only one trait, so no covariance with other traits could be calculated), but a multivariate animal model would.

Fig. S1: Black and purple are hard to distinguish. Also, and importantly, it's not clear what the panels in the two rightmost columns are plotting. It seems that these are the first and second eigenfunctions (or principal components as called here) of G; however, this does not correspond to what is in Table 1 which shows such eigenfunctions. Also, it would be convenient if figures and tables all presented the three crosses in the same order.

L 242: It seems that “stated” in L 242 should be “modified” or something like that, as the model in eq. 8 is different to that in eq. 7.

L 345: What is z?

L 270: Here and throughout the term “growth trajectory” is used to mean something that is not clear. The term growth trajectory suggests what is in Fig. S1 panels A,B,C, rather than what is in panels G,J, etc. It should be clarified what is meant by “growth trajectory”, and preferably use more precise terminology, such as eigenfunction or eigenvector.

L 284: “size” should be “breeding value of size” or something like that.

L 332: I think “would selection act” should be “were selection to act”.

L 361. Is 66 the right citation? 66 is about mute swans rather than stickleback populations mentioned in the sentence.

L 383: I think “about” should be “of”.

L 392: “ontogenetic genetic heterogeneity” is quite a mouthful. How about “genetic heterogeneity across ontogeny”.

Table 1: “loading coefficients” is not very clear. Do you mean “entries”? Also, it would be stricter to speak of eigenvalues, eigenvectors, or eigenfunctions rather than principal components.

Figure 3: Define P.

Figure S4: Transpose the arrangement of the panels to keep consistency with Fig. S3 (rows are genetic variance and columns are heritability). Again, present the crosses in the same order throughout the paper.

Review form: Reviewer 2

Recommendation

Reject – article is not of sufficient interest (we will consider a transfer to another journal)

Scientific importance: Is the manuscript an original and important contribution to its field?

Acceptable

General interest: Is the paper of sufficient general interest?

Acceptable

Quality of the paper: Is the overall quality of the paper suitable?

Acceptable

Is the length of the paper justified?

Yes

Should the paper be seen by a specialist statistical reviewer?

No

Do you have any concerns about statistical analyses in this paper? If so, please specify them explicitly in your report.

Yes

It is a condition of publication that authors make their supporting data, code and materials available - either as supplementary material or hosted in an external repository. Please rate, if applicable, the supporting data on the following criteria.

Is it accessible?

Yes

Is it clear?

Yes

Is it adequate?

Yes

Do you have any ethical concerns with this paper?

No

Comments to the Author

Main comments: In their paper, Fraimout et al. investigated the genetic architecture of body size across ontogeny in a stickleback fish. The study uses extensive data from F2-generation inter-population crosses of freshwater and marine *Pungitius pungitius*. They highlight that different QTLs contribute to the heritability of body size in different time-points. However, the heritability of body size remains constant across ontogeny. These results are very interesting in the context of predicting the response to selection in age-structured populations, and more generally, to question the relevance of classical quantitative genetic models, which assume that evolutionary parameters do not change over ontogeny. However, I am not quite convinced by the general presentation of the study, neither by the interpretation of some results. I also have a few comments about the statistical analyses used.

My first major comment is about the presentation of the study: the introduction and discussion are both focussed on parallel evolution, that is the repeated occurrence of similar phenotypic or genetic adaptation in indecently evolving populations. I understand that the stickleback has provided a very interesting study system to investigate parallel evolution, however the link between 'age-dependent genetic architecture' and 'nonparallel evolution' is far from obvious. Assuming a variation in heritability with age, nonparallel evolution would only occur if the independently evolving populations had different age-structures, or if they encountered different selection pressures. Therefore, I think the authors should probably reframe their story, and I would recommend addressing the question of the manuscript in a more general context of linking developmental processes and quantitative genetic theory. The study also needs to be more rooted in the existing literature about the ontogeny of genetic effects (a few references provided below). For instance, the authors never compared their results with previous studies reporting estimates of additive genetic variance or heritability at different ages. We need such a comparison to critically discuss the results found here.

Ontogeny of genetic effects, just a few examples: Cheverud et al. 1983 *Evolution* 37, 895-905; Gauzere et al. 2020 *Evolution*, doi: 10.1111/evo.14000 ; Lindholm et al. 2006 *Biology Letters*, doi:10.1098/rsbl.2006.0546 ; Wilson & Réale 2006 *American Naturalist*, doi: 10.1086/498138.

My second major comment is about the interpretations of the association mapping. From these analyses, the authors claim that 'different QTL control body size in different ages, and there were not global QTL which could influence growth in two or three crosses' (l. 374-376). Knowing the statistical biases of mapping, I am surprised that the authors are not discussing more critically their results. They reported a total of 11 unique QTLs significantly associated with body size variation (l. 303-304). However, they never report the estimated effect sizes, neither the total amount of genetic variation explained by these significant QTLs. I can imagine that only a small fraction of VA is explained by the detected QTLs. If so, the results most likely support a polygenic architecture of body size, and the loci underlying trait variation should be very difficult to detect significantly given their low effect sizes. For polygenic traits, we know that the results of independent mapping studies can show strong divergence. In such context, the quite consistent effect of Chr. 9 at different ontogenetic stages in the HEL x BYN and HEL x PYÖ crosses could support the evidence of one global QTL influencing body size (yet explaining a small fraction of

VA).

Concerning the statistical analyses, I understand the relevance of treating growth measurements either as a 'character state trait' or a 'function-valued trait' (l. 161-163). However, I think the authors need to provide more information about this new method implemented in the dynGP package, and it should be stated clearly that this method estimates a 'SNP-heritability' (as I understand from reading Arjas et al. *Bioinformatics* 2020). SNP-heritability and the trait heritability parameter have different definitions (de los Campos, Sorensen, Gianola *Plos Genetics* 2015), and therefore the authors probably need to compare these two estimates. I also wonder why the authors have not used the antedependence model implemented in MCMCglmm to analyse their data (see Thomson et al. *Evolution* 2017 for an example). From what I understand, this type of model is the most appropriate to deal with age-specific random effects, while accounting for the non-independence of the effects. I think the authors need to better justify their approaches.

Detailed comments:

l. 77-79: I think this is true only if the different populations have different age-structures.

l. 125-128: Why not having use the same protocol for all crosses?

l. 191-194: Probably this two-step analysis is not necessary if the authors model an antedependence structure directly in their MCMCglmm model (see Thomson et al. *Evolution* 2017 Supporting Information).

l. 195-204: Please provide more information about this new method, especially because it does not estimate the same heritability parameter than the MCMCglmm models. Then, how the analyses considering growth measurement as a 'character state trait' or a 'function-valued trait' can be compared?

l. 223-226: I understand that multivariate animal models using GRM are computationally intensive. Using animal model with pedigree data is much more efficient. Maybe using a pedigree data would then allow to use all growth measurements. Have the authors tried to compare models using GRM and pedigree data?

l. 279-281: SNP-heritability are supposed to be reported in Fig. 1, Fig S2-S4. However, the captions of these figures indicate that only the MCMCglmm results are reported. Please, indicate more clearly where these estimates can be found.

l. 292-294: how this result relates to other studies analysing the dimensionality of G matrices?

l. 303: The authors should also report the total amount of genetic variance in body size explained by these 11 QTLs.

l. 381-382: the authors need to better explain why maternal effects are not expected to be important here, as we know that juvenile traits are strongly affected by maternal effects. How the experimental design / models manage to dissociate maternal genetic effects and direct genetic effects?

Figure 1: representing CVP (or CVR) would be relevant to fully interpret the results. I think the authors also need to clearly explain in their main text that the variation in CVA is not associate with a change in h^2 because of an increase in CVP.

Review form: Reviewer 3

Recommendation

Major revision is needed (please make suggestions in comments)

Scientific importance: Is the manuscript an original and important contribution to its field?

Acceptable

General interest: Is the paper of sufficient general interest?

Good

Quality of the paper: Is the overall quality of the paper suitable?

Marginal

Is the length of the paper justified?

Yes

Should the paper be seen by a specialist statistical reviewer?

No

Do you have any concerns about statistical analyses in this paper? If so, please specify them explicitly in your report.

No

It is a condition of publication that authors make their supporting data, code and materials available - either as supplementary material or hosted in an external repository. Please rate, if applicable, the supporting data on the following criteria.

Is it accessible?

Yes

Is it clear?

Yes

Is it adequate?

No

Do you have any ethical concerns with this paper?

No

Comments to the Author

This manuscript deals with an emerging principle that is important for a fundamental understanding of phenotypic evolution: that similar phenotypic end-points can be reached with diverse genetic mechanisms. The results of this study support this notion with classical quantitative genetic and QTL analyses in an ontogenetic trait: growth of body size in nine-spined sticklebacks.

The study is impressive in the range of different techniques and modelling approaches that it combines. But this is also one of its weaknesses: many of the methods are barely explained and the results obtained barely discussed (for an example, take the Infinite Dimensional Model). As such I found it quite difficult to understand the paper.

To my mind, three changes would make this a more accessible and convincing contribution to the field:

- 1) The methods need to be explained in a lot more detail, where necessary in the Supplementary Material.
- 2) An alternative explanation for the results needs to be addressed head on: That the observed results are largely a consequence of the low statistical power that is evident in the wide confidence intervals of the estimates of heritability and CV_A. This makes the detection of any pattern difficult and it would also explain why different QTL pop up at different ages and in different crosses. With limited statistical power and many small-effect loci affecting a trait like body size, it is fairly random which loci pass the significance threshold, leading to the apparent differences in genetic architecture across ages. The authors touch on this in the Discussion, but to my mind the manuscript would benefit from a more in-depth discussion of this possibility.
- 3) There is a large body of literature on the quantitative genetics (incl. QTL) of growth in livestock. This manuscript makes few references to that literature. I think this manuscript would gain substantially if both methods and results were put into the context of what is known from livestock.

DETAILED POINTS:

Introduction: The Introduction has a conceptual 'fault line' that I found distracting. The first several paragraphs focus on the predictability of phenotypic evolution. Starting on line 76, there is a sudden and unexplained switch to considering the similarity of genetic architectures underlying the same phenotype. These two aspects are also intermingled on lines 91-96, where lines 91-93 seem to focus on the predictability of phenotypic evolution, while the following lines look at genetic architecture. To me phenotypic evolution and its genetic underpinnings are two very different perspectives that should not be mixed casually. I would suggest to focus from the outset on the similarity of the genetic architecture underlying the same phenotype, just as is done in the opening of the Discussion on lines 322-324.

L49-55: Presumably for brevity's sake, the authors use an all or none language here: is evolution predictable and will species adapt to changing environmental conditions? Although it may increase the word count, I think a more quantitative wording would be appropriate for the topic in general and this particular contribution: To what extent is evolution predictable and to what degree and how fast will species adapt to changes?

L59: See the previous comment. It would be nice to view the degree to which evolution is predictable quantitatively rather than qualitatively.

L75: Should read: '... paid to...'

L110: In what sense were these crosses independent given that in all three crosses one of the parental populations was the Helsinki population?

L134: 20'000 loci is a sizeable number of SNPs but by today's standards it is not a large number.

L155: Please explain the purpose of the Infinite Dimensional Model.

L167-170: Please make it clear that here you are considering body size a character state trait and that you ran a separate model for each age. Also, please detail what beta and gamma were made up of and what distributional assumptions you made.

L175-176: This explanation of the GRM was impenetrable for me. Please explain at a level of an average reader of the Proceedings.

L177-178: Please justify these filtering choices.

L180: Please specify and justify exactly which priors you used. mcmcglmm offers a variety of priors that are non-informative to varying degrees.

L190: Is BSA the same as y in eq. 2?

L191-194: This was impenetrable for me. Please keep an average reader of such an article in mind.

L201: Why use a different way of estimating the GRM in the two different analyses? This makes it harder to compare the results of the two analyses.

L203: This wording suggests that a regression with only an intercept and a term for sex was used, and no slope. Is this correct?

L208-211: The nature of growth trajectories introduces an autoregressive covariance structure among the epsilons in this formulation of the model. Was an autoregressive covariance structure used for the permanent environment effect?

L213: Please provide a reference for eq. 6.

L215-223: This section was impenetrable for me. Please explain in more detail what this analysis captures. Also, please clarify how the analysis here is related to the ones in the references (57,58). From my reading, references 57 and 58 use PCA to reduce the dimensionality of marker data, while here PCA is applied to the G matrix estimated from the multivariate animal models. But perhaps I misunderstand?

L241: Is reference 62 (Westfall, P.H. & Young, S.S. 1993) really appropriate here?

L251: The reference to LEP-MAP3 should be 60, and the reference for LDna should be 61. It seems that something is broken with the reference numbering and I will refrain from further commenting on mismatches.

eq. 7 and 8: Shouldn't there be a sum across loci?

L269-273: I found this hard to follow. Please explain in more detail.

L279: I did not see the SNP-heritability results in Figure 1.

L279-281: I found this wording confusing. At least qualitatively both the function-valued trait and the character state trait approach showed an increase of (C)V_A with age, but heritability to remain unchanged. That last conclusion, however, is very uncertain given the very large uncertainties in estimates of h^2 .

Figure 2: Please do not omit the estimates of the genetic correlations when they are not significant. Also, does 'multi-trait model' refer to the multivariate model on L208?

L292-293: Is this not simply the definition of the first principle component of a PCA? In other words, the fact that the first principle component explains the most variance is simply a consequence of the definition of a first principle component and not a biological result, no?

L303-308: Perhaps this textual repeat of the results in Figure 3 are not necessary?

L313: Should this be 'stages' instead of 'changes'?

L318-319: Could this be a result of low statistical power?

L325: I do not understand the focus on heritability here, given that levels of additive variation seem a more relevant measure here (see e.g. Houle 1992, Genetics). And levels of additive variation tell a different story (Figure 1): additive variation increases markedly with age in all three crosses. The reason that heritability stays the same is that the phenotypic variation also increased, which makes sense given that growth often magnifies differences over time. Incidentally, I missed a discussion of the differences between the age-related patterns of (C)V_A and heritability in the Discussion.

L337-340: The males were sampled from three different ponds, some of them quite far apart suggesting marked genetic structuring among the ponds. Would it not be a more likely explanation that the pond populations provided a more genetically diverse background, leading to the majority of the additive genetic variance to come from the segregation of pond alleles?

L340-341: This argument (that pond alleles contribute to size more strongly than marine alleles) implies dominance effects, but dominance effects were rare. Does this not suggest that dominance of pond alleles is an unlikely explanation, just as is outlined below on L342-343.

L385: They are few indisputable facts in science.

Decision letter (RSPB-2021-0954.R0)

28-Jun-2021

Dear Dr Fraimout:

I am writing to inform you that your manuscript RSPB-2021-0954 entitled "Age-dependent genetic architecture underlines similar heritability of body size in sticklebacks" has, in its current form, been rejected for publication in Proceedings B.

This action has been taken on the advice of referees, who have recommended that substantial revisions are necessary. With this in mind we would be happy to consider a resubmission, provided the comments of the referees are fully addressed. However please note that this is not a provisional acceptance.

Sincerely,
Dr Locke Rowe
mailto: proceedingsb@royalsociety.org

Associate Editor
Comments to Author:

Three referees reviewed your manuscript and differ widely in their opinions, however, they all agree that you address an important topic. There is also some overlap in their criticisms; e.g., you do not discuss previous relevant and important studies addressing similar questions. They also find that there is a lack of clarity in the focus of the manuscript. Although initially it seems focused on repeatability/predictability of evolutionary outcomes, the main and most important results address the extent to which genetic architecture is constant throughout ontogeny. Although both issues may be related, the link is not straightforward/direct so there is a need to improve the focus of your manuscript. Additionally, although your methods seem sound and appropriate for the questions at hand, there is a need to both, provide more technical details in Supplementary Information as well as to provide a more nuanced interpretation and discussion of your results. The modelling techniques used are advanced and complex so there is a need to both provide more details about them (for quantitative savvy readers) while at the same time keeping the delivery accessible to the broad Proceedings B readership. There are several other very relevant comments (e.g. referee 2 highlights the lack of estimates for effect sizes and proportion of total genetic variance explained) that you need to address. I feel all three referees provided very relevant and helpful comments that can greatly improve your manuscript if properly taken into account.

Reviewer(s)' Comments to Author:

Referee: 1

Comments to the Author(s)

This paper does a careful study of the genetic basis of body size across ontogeny in a species of stickleback fish. The paper is motivated by the interest in predicting evolution in the sense that convergent evolution of phenotype or genetic architecture results from different conditions. The authors point out that even though predictability has been found to obtain when populations have the same genetic background, predictability may not obtain when the backgrounds differ. To assess predictability in an ontogenetic context, the authors estimate the additive genetic covariation of body size across ontogeny for three sets of sticklebacks, each set having a different genetic background (the grandfather comes from a different freshwater pond in each set). Additionally, the authors assess the genetic architecture of body size across ontogeny in the three sets of sticklebacks (QTL analysis). The authors find that additive genetic variation for body size across ontogeny is substantial, that heritability of body size remains roughly unchanged across ontogeny, and that loci contributing to body size vary substantially across ontogeny and among genetic backgrounds, despite rather subtle differences in body size among genetic backgrounds.

I find the paper is rigorous in its methods and interpretations and provides interesting data regarding heterogeneous genetic architecture of body size over ontogeny depending on genetic background. I don't have any major concerns with the paper but make a few suggestions to help readers further.

The introduction is framed in terms of the predictability of evolution, but the discussion does not clearly say how the results relate to that original motivation. For instance, the conclusion revolves around how similar phenotypes may arise from different genetic architectures, and that selection on phenotypes stemming from different genetic architectures would lead to different genetic

architectures (L 395-397), the latter of which sounds like a truism. Can you conclude more directly on predictability? If not, can you say why not?

Specific comments:

Title: the word “similar” is not clear in the title (similar to what?). It would be clearer to replace it by something like “ontogenetically invariant” or something like that. Also, adding something like “Variable” as the first word in the title would clarify the point. This would yield: “Variable age-dependent genetic architecture underlines ontogenetically invariant heritability of body size in sticklebacks”.

L 24: This statement made me wonder if the authors are aware of this paper: Cheverud J. M. et al. 1983. Quantitative genetics of development: genetic correlations among age-specific trait values and the evolution of ontogeny. *Evolution*. 37, 895-905. The authors do not discuss or cite it, although that paper does very similar things to what this paper does.

L 27: I would rather say “roughly constant” or something like that rather than “constant” given the fluctuations in the mean and the wide confidence intervals.

L 54: “global change” would be better stated as “climate change” or “global climate change”.

L 68,69: I don’t think the authors mean “positive function” here (any probability is a positive function whenever it is non-zero). Do you mean “increasing function”?

L 67-69: If “positive function” means “increasing function”, then this statement seems to say that gene flow (i.e., different pools) increases effective population size and so predictability. This seems to contradict the immediately previous statement in L 64-66 that evolution is less predictable with variation from different pools.

L75: I think it should be “attention has been paid to” rather than “on”.

L 96: I think the statement that the evolutionary outcome would be unpredictable is easily misunderstood if taken out of context. Here predictability has a narrow meaning (i.e., obtaining the same outcome from different conditions), but predictability in common speech has a broader meaning. So, stating that the evolutionary outcome is unpredictable is easily understood in the broader meaning (i.e., that you can’t predict the outcome, but in principle you could if you had a lot more information about the system). Anyway, this is just to say that it would be good to add a parenthesis here reminding the reader that predictability here has a narrow sense (i.e., repeatability).

L 153: t_0 is not the estimated length at age 0, but “a value used to calculate size when age is zero” (phrasing taken from Wikipedia, Von Bertalanffy function article).

L 169: What are the X and Y measuring? Age and gene content, respectively? Some more information on what these matrices contain is needed here.

L175: “additive coefficients” of what? Are these regression coefficients of phenotype on SNP’s, or something like that? Again, some more specification of W is needed.

L 183: Rather than “each size”, do you mean just “size”?

L 197: I think the statement “Contrary to the univariate animal model approach” is not helpful here. A univariate animal model would not even allow for the calculation of genetic covariances (there is only one trait, so no covariance with other traits could be calculated), but a multivariate animal model would.

Fig. S1: Black and purple are hard to distinguish. Also, and importantly, it's not clear what the panels in the two rightmost columns are plotting. It seems that these are the first and second eigenfunctions (or principal components as called here) of G ; however, this does not correspond to what is in Table 1 which shows such eigenfunctions. Also, it would be convenient if figures and tables all presented the three crosses in the same order.

L 242: It seems that "stated" in L 242 should be "modified" or something like that, as the model in eq. 8 is different to that in eq. 7.

L 345: What is z ?

L 270: Here and throughout the term "growth trajectory" is used to mean something that is not clear. The term growth trajectory suggests what is in Fig. S1 panels A,B,C, rather than what is in panels G,J, etc. It should be clarified what is meant by "growth trajectory", and preferably use more precise terminology, such as eigenfunction or eigenvector.

L 284: "size" should be "breeding value of size" or something like that.

L 332: I think "would selection act" should be "were selection to act".

L 361. Is 66 the right citation? 66 is about mute swans rather than stickleback populations mentioned in the sentence.

L 383: I think "about" should be "of".

L 392: "ontogenetic genetic heterogeneity" is quite a mouthful. How about "genetic heterogeneity across ontogeny".

Table 1: "loading coefficients" is not very clear. Do you mean "entries"? Also, it would be stricter to speak of eigenvalues, eigenvectors, or eigenfunctions rather than principal components.

Figure 3: Define P .

Figure S4: Transpose the arrangement of the panels to keep consistency with Fig. S3 (rows are genetic variance and columns are heritability). Again, present the crosses in the same order throughout the paper.

Referee: 2

Comments to the Author(s)

Main comments: In their paper, Fraimout et al. investigated the genetic architecture of body size across ontogeny in a stickleback fish. The study uses extensive data from F2-generation inter-population crosses of freshwater and marine *Pungitius pungitius*. They highlight that different QTLs contribute to the heritability of body size in different time-points. However, the heritability of body size remains constant across ontogeny. These results are very interesting in the context of predicting the response to selection in age-structured populations, and more generally, to question the relevance of classical quantitative genetic models, which assume that evolutionary parameters do not change over ontogeny. However, I am not quite convinced by the general presentation of the study, neither by the interpretation of some results. I also have a few comments about the statistical analyses used.

My first major comment is about the presentation of the study: the introduction and discussion are both focussed on parallel evolution, that is the repeated occurrence of similar phenotypic or genetic adaptation in indecently evolving populations. I understand that the stickleback has provided a very interesting study system to investigate parallel evolution, however the link between 'age-dependent genetic architecture' and 'nonparallel evolution' is far from obvious.

Assuming a variation in heritability with age, nonparallel evolution would only occur if the independently evolving populations had different age-structures, or if they encountered different selection pressures. Therefore, I think the authors should probably reframe their story, and I would recommend addressing the question of the manuscript in a more general context of linking developmental processes and quantitative genetic theory. The study also needs to be more rooted in the existing literature about the ontogeny of genetic effects (a few references provided below). For instance, the authors never compared their results with previous studies reporting estimates of additive genetic variance or heritability at different ages. We need such a comparison to critically discuss the results found here.

Ontogeny of genetic effects, just a few examples: Cheverud et al. 1983 *Evolution* 37, 895-905; Gauzere et al. 2020 *Evolution*, doi: 10.1111/evo.14000 ; Lindholm et al. 2006 *Biology Letters*, doi:10.1098/rsbl.2006.0546 ; Wilson & Réale 2006 *American Naturalist*, doi: 10.1086/498138.

My second major comment is about the interpretations of the association mapping. From these analyses, the authors claim that 'different QTL control body size in different ages, and there were not global QTL which could influence growth in two or three crosses' (l. 374-376). Knowing the statistical biases of mapping, I am surprised that the authors are not discussing more critically their results. They reported a total of 11 unique QTLs significantly associated with body size variation (l. 303-304). However, they never report the estimated effect sizes, neither the total amount of genetic variation explained by these significant QTLs. I can imagine that only a small fraction of VA is explained by the detected QTLs. If so, the results most likely support a polygenic architecture of body size, and the loci underlying trait variation should be very difficult to detect significantly given their low effect sizes. For polygenic traits, we know that the results of independent mapping studies can show strong divergence. In such context, the quite consistent effect of Chr. 9 at different ontogenetic stages in the HEL x BYN and HEL x PYÖ crosses could support the evidence of one global QTL influencing body size (yet explaining a small fraction of VA).

Concerning the statistical analyses, I understand the relevance of treating growth measurements either as a 'character state trait' or a 'function-valued trait' (l. 161-163). However, I think the authors need to provide more information about this new method implemented in the dynGP package, and it should be stated clearly that this method estimates a 'SNP-heritability' (as I understand from reading Arjas et al. *Bioinformatics* 2020). SNP-heritability and the trait heritability parameter have different definitions (de los Campos, Sorensen, Gianola *Plos Genetics* 2015), and therefore the authors probably need to compare these two estimates. I also wonder why the authors have not used the antedependence model implemented in MCMCglmm to analyse their data (see Thomson et al. *Evolution* 2017 for an example). From what I understand, this type of model is the most appropriate to deal with age-specific random effects, while accounting for the non-independence of the effects. I think the authors need to better justify their approaches.

Detailed comments:

l. 77-79: I think this is true only if the different populations have different age-structures.

l. 125-128: Why not having use the same protocol for all crosses?

l. 191-194: Probably this two-step analysis is not necessary if the authors model an antedependence structure directly in their MCMCglmm model (see Thomson et al. *Evolution* 2017 Supporting Information).

l. 195-204: Please provide more information about this new method, especially because it does not estimate the same heritability parameter than the MCMCglmm models. Then, how the analyses considering growth measurement as a 'character state trait' or a 'function-valued trait' can be compared?

l. 223-226: I understand that multivariate animal models using GRM are computationally intensive. Using animal model with pedigree data is much more efficient. Maybe using a pedigree data would then allow to use all growth measurements. Have the authors tried to compare models using GRM and pedigree data?

l. 279-281: SNP-heritability are supposed to be reported in Fig. 1, Fig S2-S4. However, the captions of these figures indicate that only the MCMCglmm results are reported. Please, indicate more clearly where these estimates can be found.

l. 292-294: how this result relates to other studies analysing the dimensionality of G matrices?

l. 303: The authors should also report the total amount of genetic variance in body size explained by these 11 QTLs.

l. 381-382: the authors need to better explain why maternal effects are not expected to be important here, as we know that juvenile traits are strongly affected by maternal effects. How the experimental design / models manage to dissociate maternal genetic effects and direct genetic effects?

Figure 1: representing CVP (or CVR) would be relevant to fully interpret the results. I think the authors also need to clearly explain in their main text that the variation in CVA is not associated with a change in h^2 because of an increase in CVP.

Referee: 3

Comments to the Author(s)

This manuscript deals with an emerging principle that is important for a fundamental understanding of phenotypic evolution: that similar phenotypic end-points can be reached with diverse genetic mechanisms. The results of this study support this notion with classical quantitative genetic and QTL analyses in an ontogenetic trait: growth of body size in nine-spined sticklebacks.

The study is impressive in the range of different techniques and modelling approaches that it combines. But this is also one of its weaknesses: many of the methods are barely explained and the results obtained barely discussed (for an example, take the Infinite Dimensional Model). As such I found it quite difficult to understand the paper.

To my mind, three changes would make this a more accessible and convincing contribution to the field:

1) The methods need to be explained in a lot more detail, where necessary in the Supplementary Material.

2) An alternative explanation for the results needs to be addressed head on: That the observed results are largely a consequence of the low statistical power that is evident in the wide confidence intervals of the estimates of heritability and CV_A. This makes the detection of any pattern difficult and it would also explain why different QTL pop up at different ages and in different crosses. With limited statistical power and many small-effect loci affecting a trait like body size, it is fairly random which loci pass the significance threshold, leading to the apparent differences in genetic architecture across ages. The authors touch on this in the Discussion, but to my mind the manuscript would benefit from a more in-depth discussion of this possibility.

3) There is a large body of literature on the quantitative genetics (incl. QTL) of growth in livestock. This manuscript makes few references to that literature. I think this manuscript would gain substantially if both methods and results were put into the context of what is known from livestock.

DETAILED POINTS:

Introduction: The Introduction has a conceptual 'fault line' that I found distracting. The first several paragraphs focus on the predictability of phenotypic evolution. Starting on line 76, there is a sudden and unexplained switch to considering the similarity of genetic architectures underlying the same phenotype. These two aspects are also intermingled on lines 91-96, where lines 91-93 seem to focus on the predictability of phenotypic evolution, while the following lines look at genetic architecture. To me phenotypic evolution and its genetic underpinnings are two very different perspectives that should not be mixed casually. I would suggest to focus from the outset on the similarity of the genetic architecture underlying the same phenotype, just as is done in the opening of the Discussion on lines 322-324.

L49-55: Presumably for brevity's sake, the authors use an all or none language here: is evolution predictable and will species adapt to changing environmental conditions? Although it may increase the word count, I think a more quantitative wording would be appropriate for the topic in general and this particular contribution: To what extent is evolution predictable and to what degree and how fast will species adapt to changes?

L59: See the previous comment. It would be nice to view the degree to which evolution is predictable quantitatively rather than qualitatively.

L75: Should read: '... paid to...'

L110: In what sense were these crosses independent given that in all three crosses one of the parental populations was the Helsinki population?

L134: 20'000 loci is a sizeable number of SNPs but by today's standards it is not a large number.

L155: Please explain the purpose of the Infinite Dimensional Model.

L167-170: Please make it clear that here you are considering body size a character state trait and that you ran a separate model for each age. Also, please detail what beta and gamma were made up of and what distributional assumptions you made.

L175-176: This explanation of the GRM was impenetrable for me. Please explain at a level of an average reader of the Proceedings.

L177-178: Please justify these filtering choices.

L180: Please specify and justify exactly which priors you used. mcmcglmm offers a variety of priors that are non-informative to varying degrees.

L190: Is BSA the same as y in eq. 2?

L191-194: This was impenetrable for me. Please keep an average reader of such an article in mind.

L201: Why use a different way of estimating the GRM in the two different analyses? This makes it harder to compare the results of the two analyses.

L203: This wording suggests that a regression with only an intercept and a term for sex was used, and no slope. Is this correct?

L208-211: The nature of growth trajectories introduces an autoregressive covariance structure among the epsilons in this formulation of the model. Was an autoregressive covariance structure used for the permanent environment effect?

L213: Please provide a reference for eq. 6.

L215-223: This section was impenetrable for me. Please explain in more detail what this analysis captures. Also, please clarify how the analysis here is related to the ones in the references (57,58). From my reading, references 57 and 58 use PCA to reduce the dimensionality of marker data, while here PCA is applied to the G matrix estimated from the multivariate animal models. But perhaps I misunderstand?

L241: Is reference 62 (Westfall, P.H. & Young, S.S. 1993) really appropriate here?

L251: The reference to LEP-MAP3 should be 60, and the reference for LDna should be 61. It seems that something is broken with the reference numbering and I will refrain from further commenting on mismatches.

eq. 7 and 8: Shouldn't there be a sum across loci?

L269-273: I found this hard to follow. Please explain in more detail.

L279: I did not see the SNP-heritability results in Figure 1.

L279-281: I found this wording confusing. At least qualitatively both the function-valued trait and the character state trait approach showed an increase of (C)V_A with age, but heritability to remain unchanged. That last conclusion, however, is very uncertain given the very large uncertainties in estimates of h^2 .

Figure 2: Please do not omit the estimates of the genetic correlations when they are not significant. Also, does 'multi-trait model' refer to the multivariate model on L208?

L292-293: Is this not simply the definition of the first principle component of a PCA? In other words, the fact that the first principle component explains the most variance is simply a consequence of the definition of a first principle component and not a biological result, no?

L303-308: Perhaps this textual repeat of the results in Figure 3 are not necessary?

L313: Should this be 'stages' instead of 'changes'?

L318-319: Could this be a result of low statistical power?

L325: I do not understand the focus on heritability here, given that levels of additive variation seem a more relevant measure here (see e.g. Houle 1992, Genetics). And levels of additive variation tell a different story (Figure 1): additive variation increases markedly with age in all three crosses. The reason that heritability stays the same is that the phenotypic variation also increased, which makes sense given that growth often magnifies differences over time. Incidentally, I missed a discussion of the differences between the age-related patterns of (C)V_A and heritability in the Discussion.

L337-340: The males were sampled from three different ponds, some of them quite far apart suggesting marked genetic structuring among the ponds. Would it not be a more likely explanation that the pond populations provided a more genetically diverse background, leading to the majority of the additive genetic variance to come from the segregation of pond alleles?

L340-341: This argument (that pond alleles contribute to size more strongly than marine alleles) implies dominance effects, but dominance effects were rare. Does this not suggest that dominance of pond alleles is an unlikely explanation, just as is outlined below on L342-343.

L385: They are few indisputable facts in science.

Author's Response to Decision Letter for (RSPB-2021-0954.R0)

See Appendix A.

RSPB-2022-0352.R0

Review form: Reviewer 2

Recommendation

Accept as is

Scientific importance: Is the manuscript an original and important contribution to its field?

Excellent

General interest: Is the paper of sufficient general interest?

Excellent

Quality of the paper: Is the overall quality of the paper suitable?

Excellent

Is the length of the paper justified?

Yes

Should the paper be seen by a specialist statistical reviewer?

No

Do you have any concerns about statistical analyses in this paper? If so, please specify them explicitly in your report.

No

It is a condition of publication that authors make their supporting data, code and materials available - either as supplementary material or hosted in an external repository. Please rate, if applicable, the supporting data on the following criteria.

Is it accessible?

Yes

Is it clear?

Yes

Is it adequate?

Yes

Do you have any ethical concerns with this paper?

No

Comments to the Author

I really appreciate the effort the authors took to improve their presentation. They have addressed my previous comments thoroughly, and it seems they did the same with the comments from

other reviewers. Therefore, I have no further comments.

Review form: Reviewer 3

Recommendation

Major revision is needed (please make suggestions in comments)

Scientific importance: Is the manuscript an original and important contribution to its field?

Good

General interest: Is the paper of sufficient general interest?

Good

Quality of the paper: Is the overall quality of the paper suitable?

Good

Is the length of the paper justified?

Yes

Should the paper be seen by a specialist statistical reviewer?

No

Do you have any concerns about statistical analyses in this paper? If so, please specify them explicitly in your report.

No

It is a condition of publication that authors make their supporting data, code and materials available - either as supplementary material or hosted in an external repository. Please rate, if applicable, the supporting data on the following criteria.

Is it accessible?

Yes

Is it clear?

Yes

Is it adequate?

Yes

Do you have any ethical concerns with this paper?

No

Comments to the Author

The authors have substantially revised the manuscript. It is now considerably easier to read and understand and I thank the authors for this effort.

I have two main comments on the revised version of the manuscript.

1) The authors failed to address one important issue in the revision. The manuscript continues to ignore the large body of literature on QTLs of growth in farm animals. Studies of farmed animals (including fish) over the past nearly 20 years have shown that QTLs for growth are often age-dependent, just as reported in the present study. Thus, contrary to claims in the present

manuscript (e.g. lines 35-36, 130-142, 584) the fact that the genetic architecture of growth is ontogenetically heterogeneous is empirically well-established and not novel, not even in fish.

To my mind, this does not diminish the value of the present study. This manuscript provides a nice illustration of this general principle in sticklebacks. But the authors need to acknowledge the large literature that already exists on this topic, and they need to put their results into the context of this existing knowledge in other species rather than claiming novelty.

Here are just a few of many more studies on age-dependent QTLs in farm animals:

Carlborg, O., Kerje, S., Schütz, K., Jacobsson, L., Jensen, P., & Andersson, L. (2003). A global search reveals epistatic interaction between QTL for early growth in the chicken. *Genome research*, 13(3), 413-421. <https://doi.org/10.1101/gr.528003>

Hadjipavlou, G. and Bishop, S.C. (2009), Age-dependent quantitative trait loci affecting growth traits in Scottish Blackface sheep. *Animal Genetics*, 40: 165-175. <https://doi.org/10.1111/j.1365-2052.2008.01814.x>

Podisi, B.K., Knott, S.A., Burt, D.W. et al. Comparative analysis of quantitative trait loci for body weight, growth rate and growth curve parameters from 3 to 72 weeks of age in female chickens of a broiler-layer cross. *BMC Genet* 14, 22 (2013). <https://doi.org/10.1186/1471-2156-14-22>

Miyako Kodama, Jeffrey J. Hard, Kerry A. Naish. 2018. Mapping of quantitative trait loci for temporal growth and age at maturity in coho salmon: Evidence for genotype-by-sex interactions. *Marine Genomics* (38): 33-44. <https://doi.org/10.1016/j.margen.2017.07.004>.

2) Perhaps I made a mistake somewhere but the files I could download did not include the figures and tables. Hence, I cannot comment on the figures and tables and the result presented therein.

Detailed Points:

- Abstract: The Abstract would profit from more substantial changes in light of the other changes in the manuscript.

- L. 323: 'Gaussian' should perhaps be capitalised?

- L. 462: Should read 'grandparental'.

- L. 578: I wasn't sure how to reconcile the statement here, that phenotypic growth trajectories are very similar, with Figure S1 and with the significant additive genetic variation found in this study. The latter two suggest to me that there was substantial variation in growth trajectories, which goes nicely with the QTL results. Thus, to my mind and contrary to the wording here, there is no fundamental contradiction between genetic and phenotypic patterns.

Decision letter (RSPB-2022-0352.R0)

04-Apr-2022

Dear Dr Fraimout:

Your manuscript has now been peer reviewed and the reviews have been assessed by an Associate Editor. The reviewers' comments (not including confidential comments to the Editor)

and the comments from the Associate Editor are included at the end of this email for your reference. As you will see, the reviewers and the Editors have raised some concerns with your manuscript and we would like to invite you to revise your manuscript to address them.

When submitting your revision please upload a file under "Response to Referees" - in the "File Upload" section. This should document, point by point, how you have responded to the reviewers' and Editors' comments, and the adjustments you have made to the manuscript. We also require a copy of the revised manuscript showing track changes to be uploaded.

Research ethics:

Use of animals and field studies:

It is a condition of publication that data supporting your paper are made available either in the electronic supplementary material. Authors must complete the 'data accessibility' section in the submission system. This should list the database and accession number for all data from the article that has been made publicly available, for instance:

NB. From April 1 2013, peer reviewed articles based on research funded wholly or partly by RCUK must include, if applicable, a statement on how the underlying research materials – such as data, samples or models – can be accessed.

[http://datadryad.org/submit?journalID=RSPB&manu=\(Document not available\)](http://datadryad.org/submit?journalID=RSPB&manu=(Document not available)) which will

take you to your unique entry in the Dryad repository. If you have already submitted your data to dryad you can make any necessary revisions to your dataset by following the above link.

Please include the Dryad DOI in the Data Accessibility section and reference in the paper's bibliography.

Please see our Data Sharing Policies (<https://royalsociety.org/journals/authors/author-guidelines/>).

Please submit a copy of your revised paper within three weeks. If we do not hear from you within this time your manuscript will be rejected. If you are unable to meet this deadline please let us know as soon as possible, as we may be able to grant a short extension.

Thank you for submitting your manuscript to *Proceedings B*; we look forward to receiving your revision. If you have any questions at all, please do not hesitate to get in touch.

Best wishes,
Dr Locke Rowe
mailto:proceedingsb@royalsociety.org

Associate Editor
Comments to Author:

Two of the referees who evaluated your original manuscript have now provided comments for the revised version. As you will see, they are very positive but referee 2 makes a very important point that needs to be addressed. You do need to acknowledge the fact that numerous existing studies focused on farm animals have already demonstrated empirically that the genetic architecture of growth is ontogenically heterogeneous in several species.

Your study would be made even more valuable by discussing the existing reported evidence for ontogenic heterogeneity in genetic architecture of size/growth and contrasting it with your results as it represents a very thorough and rigorous demonstration that this process can be important for local adaptation, which goes well beyond what other studies have done.

Reviewer(s)' Comments to Author:

Referee: 2

Comments to the Author(s).

I really appreciate the effort the authors took to improve their presentation. They have addressed my previous comments thoroughly, and it seems they did the same with the comments from other reviewers. Therefore, I have no further comments.

Referee: 3

Comments to the Author(s).

The authors have substantially revised the manuscript. It is now considerably easier to read and understand and I thank the authors for this effort.

I have two main comments on the revised version of the manuscript.

1) The authors failed to address one important issue in the revision. The manuscript continues to ignore the large body of literature on QTLs of growth in farm animals. Studies of farmed animals (including fish) over the past nearly 20 years have shown that QTLs for growth are often age-dependent, just as reported in the present study. Thus, contrary to claims in the present manuscript (e.g. lines 35-36, 130-142, 584) the fact that the genetic architecture of growth is ontogenetically heterogeneous is empirically well-established and not novel, not even in fish.

To my mind, this does not diminish the value of the present study. This manuscript provides a nice illustration of this general principle in sticklebacks. But the authors need to acknowledge the large literature that already exists on this topic, and they need to put their results into the context of this existing knowledge in other species rather than claiming novelty.

Here are just a few of many more studies on age-dependent QTLs in farm animals:

Carlborg, O., Kerje, S., Schütz, K., Jacobsson, L., Jensen, P., & Andersson, L. (2003). A global search reveals epistatic interaction between QTL for early growth in the chicken. *Genome research*, 13(3), 413–421. <https://doi.org/10.1101/gr.528003>

Hadjipavlou, G. and Bishop, S.C. (2009), Age-dependent quantitative trait loci affecting growth traits in Scottish Blackface sheep. *Animal Genetics*, 40: 165-175. <https://doi.org/10.1111/j.1365-2052.2008.01814.x>

Podisi, B.K., Knott, S.A., Burt, D.W. et al. Comparative analysis of quantitative trait loci for body weight, growth rate and growth curve parameters from 3 to 72 weeks of age in female chickens of a broiler-layer cross. *BMC Genet* 14, 22 (2013). <https://doi.org/10.1186/1471-2156-14-22>

Miyako Kodama, Jeffrey J. Hard, Kerry A. Naish. 2018. Mapping of quantitative trait loci for temporal growth and age at maturity in coho salmon: Evidence for genotype-by-sex interactions. *Marine Genomics* (38): 33-44. <https://doi.org/10.1016/j.margen.2017.07.004>.

2) Perhaps I made a mistake somewhere but the files I could download did not include the figures and tables. Hence, I cannot comment on the figures and tables and the result presented therein.

Detailed Points:

- Abstract: The Abstract would profit from more substantial changes in light of the other changes in the manuscript.

- L. 323: 'Gaussian' should perhaps be capitalised?

- L. 462: Should read 'grandparental'.

- L. 578: I wasn't sure how to reconcile the statement here, that phenotypic growth trajectories are very similar, with Figure S1 and with the significant additive genetic variation found in this study. The latter two suggest to me that there was substantial variation in growth trajectories, which goes nicely with the QTL results. Thus, to my mind and contrary to the wording here, there is no fundamental contradiction between genetic and phenotypic patterns.

Author's Response to Decision Letter for (RSPB-2022-0352.R0)

See Appendix B.

Decision letter (RSPB-2022-0352.R1)

19-Apr-2022

Dear Dr Fraimout

I am pleased to inform you that your manuscript entitled "Age-dependent genetic architecture across ontogeny of body size in sticklebacks" has been accepted for publication in Proceedings B.

Data Accessibility section

Open Access

Paper charges

Sincerely,
Dr Locke Rowe
Editor, Proceedings B
mailto: proceedingsb@royalsociety.org

Associate Editor:

Comments to Author:

Thanks for incorporating the additional comments made by referee 3.

Appendix A

Associate Editor

Three referees reviewed your manuscript and differ widely in their opinions, however, they all agree that you address an important topic. There is also some overlap in their criticisms; e.g., you do not discuss previous relevant and important studies addressing similar questions. They also find that there is a lack of clarity in the focus of the manuscript. Although initially it seems focused on repeatability/predictability of evolutionary outcomes, the main and most important results address the extent to which genetic architecture is constant throughout ontogeny. Although both issues may be related, the link is not straightforward/direct so there is a need to improve the focus of your manuscript.

We thank the Editor and all Referees for thinking that our study addresses important questions. We are particularly grateful for the astute comments received on our manuscript. We acknowledge that the previous version lacked some focus on the scope of our study and that consequently, some important work in the field was left un-discussed and un-cited.

With this revised manuscript, we made a particular effort in rewriting the text to better frame our study. Particularly, we now clearly state the conceptual idea behind our study that age-specific heterogeneity in genetic architecture could promote non-parallel evolution by increasing the heterogeneity of genetic material available for selection to act on, thus reducing the probability of parallel evolution (*viz.* reducing the predictability of evolution). This is now clearly outlined in our revised introduction along with clear hypothesis and objective to be tested in the manuscript.

Additionally, although your methods seem sound and appropriate for the questions at hand, there is a need to both, provide more technical details in Supplementary Information as well as to provide a more nuanced interpretation and discussion of your results. The modelling techniques used are advanced and complex so there is a need to both provide more details about them (for quantitative savvy readers) while at the same time keeping the delivery accessible to the broad Proceedings B readership.

In line with our previous response, we have made a particular effort in rewriting the Methods section of our manuscript following all of the Referees comments. The main text in the revised Methods section now provides clear and accessible information about our different analyses.

Furthermore, we provide a detailed *Supplementary Methods* section as a companion to this revised Methods section, which gives additional technical details on all analyses performed in the manuscript.

There are several other very relevant comments (e.g., referee 2 highlights the lack of estimates for effect sizes and proportion of total genetic variance explained) that you need to address. I feel all three referees provided very relevant and helpful comments that can greatly improve your manuscript if properly taken into account.

We have now addressed all comments and suggestions from all Referees which were, indeed,

very relevant. The major revisions added to our manuscript were (but not limited to: see our detailed responses below):

- Re-writing of the Introduction with an emphasis on clearly explaining the link between age-specific genetic architecture and the likelihood of parallel evolution.
- Re-writing of the Methods section with the aim of providing more accessible information about our different analyses while providing detailed statistical and methodological elements in the new *Supplementary Methods* section.
- Addition of effect sizes and PVE estimates for all QTL reported.
- Re-writing of the Discussion with a particular effort to discuss key elements of our results such as statistical power limitations, the polygenic nature of our study trait and the link between our results and the study of parallel evolution.
- An improved set of citations and references throughout the manuscript.

We also note that along with these substantial revisions, and due to the length limits for manuscripts, we have decided to move Figure 1 to the *Supplementary material* as some of the information of this Figure (i.e. heritability estimates) was redundant with Figure 2 (now Figure 1). For the same reason, we had to move several sections of the *Material & Methods* to *Supplementary material*. These include: details about the rearing conditions of the individuals, detailed description of SNP genotyping and mathematical derivations of the QTL models.

Please find below our point-by-point responses to all comments received (in blue text). We have also numbered each Referee's comment for ease of response. All line-numbering references correspond to the revised version of the manuscript without track changes.

We sincerely hope that this new revision will be satisfactory to the Editor and all the Referees.

Referee 1

#1. This paper does a careful study of the genetic basis of body size across ontogeny in a species of stickleback fish. The paper is motivated by the interest in predicting evolution in the sense that convergent evolution of phenotype or genetic architecture results from different conditions. The authors point out that even though predictability has been found to obtain when populations have the same genetic background, predictability may not obtain when the backgrounds differ. To assess predictability in an ontogenetic context, the authors estimate the additive genetic covariation of body size across ontogeny for three sets of sticklebacks, each set having a different genetic background (the grandfather comes from a different freshwater pond in each set). Additionally, the authors assess the genetic architecture of body size across ontogeny in the three sets of sticklebacks (QTL analysis). The authors find that additive genetic variation for body size across ontogeny is substantial, that heritability of body size remains roughly unchanged across ontogeny, and that loci contributing to body size vary substantially across ontogeny and among genetic backgrounds, despite rather subtle differences in body size among genetic backgrounds.

I find the paper is rigorous in its methods and interpretations and provides interesting data regarding heterogenous genetic architecture of body size over ontogeny depending on genetic

background. I don't have any major concerns with the paper but make a few suggestions to help readers further.

We thank the Referee for this positive comment on our manuscript and hope that our responses below will be satisfactory.

#2. The introduction is framed in terms of the predictability of evolution, but the discussion does not clearly say how the results relate to that original motivation.

We acknowledge that our initial introduction was unclear in our intention to link the genetic architecture of ontogeny to parallel evolution.

We propose that age-specific genetic architecture could act as a potential source of non-parallelism if selection were to act at different ages. Indeed, the predictability of evolution can be defined as the probability of parallel evolution to occur (i.e., that similar phenotypes would be underlined by similar genetic architectures).

Following this definition, any process that would increase the heterogeneity of genetic material available for selection would therefore reduce the probability of parallel evolution and, ultimately, the predictability of evolution. We propose that age-specific genetic architecture could be such a process, by increasing the heterogeneity of genetic architecture underlying similar phenotypes in natural populations.

We have now substantially revised our Introduction (L76-82) and Discussion (L388-396) sections in line with this response and following the Referee's comment, which will hopefully give a better framing to our study.

#3. For instance, the conclusion revolves around how similar phenotypes may arise from different genetic architectures, and that selection on phenotypes stemming from different genetic architectures would lead to different genetic architectures (L 395-397), the latter of which sounds like a truism. Can you conclude more directly on predictability? If not, can you say why not?

As outlined in our previous response to comment #2, we discuss the possibility that heterogeneity in genetic architecture across ontogeny would lead to non-parallel evolution if selection acting at different ages (thus, on different genetic architecture) would result in similar phenotypic outcome.

Our results support this conceptual idea in two ways: i) ample genetic variance and heritability across development would allow the selected trait (in our case, body size) to respond to age-specific selection and ii) different age-specific QTL would be selected for by selection taking place at different ages.

Regarding the notion of predictability, the point raised by the Referee in this comment is particularly relevant to our study and as mentioned in our response above, we have added text in the Introduction (see L76-82), as well as a concluding paragraph to our revised Discussion

(L388-396) that directly addresses the predictability of evolution and how our results integrate with this notion.

Specifically, we explain how age-specific heterogeneity in genetic architecture can decrease the predictability of evolution by decreasing the probability of parallel evolution.

Specific comments:

#4. Title: the word “similar” is not clear in the title (similar to what?). It would be clearer to replace it by something like “ontogenetically invariant” or something like that. Also, adding something like “Variable” as the first word in the title would clarify the point. This would yield: “Variable age-dependent genetic architecture underlines ontogenetically invariant heritability of body size in sticklebacks”.

We thank the Referee for this helpful comment on our title. In response to comments from all Referees, we have decided to tone down the focus on the constancy of heritability throughout the manuscript (and see our response to comment #6 below and response to Referee 3’s comment #2). We also agree with the Referee the word “similar” was not clear in the previous title and decided to use the word “across” to specify that we are referring to all stages of ontogeny.

Therefore, we decided to simplify the title and focus on the main finding of the study: “Age-specific genetic architecture across ontogeny of body size in sticklebacks”.

#5. L 24: This statement made me wonder if the authors are aware of this paper: Cheverud J. M. et al. 1983. Quantitative genetics of development: genetic correlations among age-specific trait values and the evolution of ontogeny. *Evolution*. 37, 895-905. The authors do not discuss or cite it, although that paper does very similar things to what this paper does.

The Referee makes a very good point here and we are aware of the Cheverud *et al.* (1983) study which we extensively read in the process of setting up our own study. The lack of citation was simply a mistake on our part, which has now been corrected (see reference [27]).

#6. L 27: I would rather say “roughly constant” or something like that rather than “constant” given the fluctuations in the mean and the wide confidence intervals.

We have changed the sentence in line with the Referee’s suggestion, which now reads “[...] *remains approximately constant across ontogeny* [...]”. Furthermore, we have toned down the parts of the text relating to the constancy of heritability throughout the manuscript also in response to the other comments received.

#7. L 54: “global change” would be better stated as “climate change” or “global climate change”.

We have reworded this following the Referee’s suggestions which now reads “*global environmental change*”.

#8. L 68,69: I don't think the authors mean "positive function" here (any probability is a positive function whenever it is non-zero). Do you mean "increasing function"?

This sentence has been removed in the process of re-writing of the Introduction.

#9. L 67-69: If "positive function" means "increasing function", then this statement seems to say that gene flow (i.e., different pools) increases effective population size and so predictability. This seems to contradict the immediately previous statement in L 64-66 that evolution is less predictable with variation from different pools.

This sentence has been removed in the process of re-writing of the Introduction

#10. L75: I think it should be "attention has been paid to" rather than "on".

This has been corrected.

#11. L 96: I think the statement that the evolutionary outcome would be unpredictable is easily misunderstood if taken out of context. Here predictability has a narrow meaning (i.e., obtaining the same outcome from different conditions), but predictability in common speech has a broader meaning. So, stating that the evolutionary outcome is unpredictable is easily understood in the broader meaning (i.e., that you can't predict the outcome, but in principle you could if you had a lot more information about the system). Anyway, this is just to say that it would be good to add a parenthesis here reminding the reader that predictability here has a narrow sense (i.e., repeatability).

We understand the point raised by the Referee but perhaps our writing led to a misunderstanding here. In this part of the text (now L96-99 in the revised version), we do indeed mean that selection acting on age-specific genetic architecture would result in unpredictable genetic outcome in the broader meaning (to use the Referee's term).

In line with our previous response to the Referee's comment #3 on the matter of predictability, we added text in the Introduction (L76-82) and Discussion (L388-396) that should clarify this point as we clearly define the predictability of evolution with regards to the probability of parallel evolution.

#12. L 153: t_0 is not the estimated length at age 0, but "a value used to calculate size when age is zero" (phrasing taken from Wikipedia, Von Bertalanffy function article).

We found throughout our literature review that the term " t_0 " was in fact defined differently between studies. This probably comes from the fact that "age 0" is in itself a rather abstract concept. We propose the following correction in our revised manuscript and hope it will be satisfactory to the Referee: " t_0 is the estimated *hypothetical* length at age $t = 0$ "

#13. L 169: What are the X and Y measuring? Age and gene content, respectively? Some more information on what these matrices contain is needed here.

We apologize for the misunderstanding but we are not sure if the Referee is referring here to the design matrices X and Z , as equation (2) does not contain a capital Y .

If so, X and Z correspond respectively to the fixed and random effects the design matrices of the linear regression and specifically contain the matrix of phenotypic observations (the individual values of body size at one age, along with the sex information of the individuals included in β) and the matrix of relatedness coefficient (the GRM).

If the Referee is referring to the lower-case 'y' of the equation, it is the vector of predicted values from the linear model.

In either case, all of the above terms are now defined as in our response above in the main text.

#14. L175: “additive coefficients” of what? Are these regression coefficients of phenotype on SNP's, or something like that? Again, some more specification of W is needed.

Here, “additive coefficients” refers to the coefficient of gene additivity coded in each genotype. Specifically, they correspond to the 0, 1 and 2 values for the AA, AB and BB genotypes, respectively. However, this text has been removed in the process of revision.

Following the Referee's comment as well as Referee 2's and 3's comments, we have added a detailed *Supplementary methods* section to our manuscript which explains all the methods employed in our study. In this section we give a more detailed description of the Genomic Relationship Matrix.

#15. L 183: Rather than “each size”, do you mean just “size”?

Yes, this has been corrected.

#16. L 197: I think the statement “Contrary to the univariate animal model approach” is not helpful here. A univariate animal model would not even allow for the calculation of genetic covariances (there is only one trait, so no covariance with other traits could be calculated), but a multivariate animal model would.

We agree with the Referee and decided to remove this statement from the text to avoid confusion.

#17. Fig. S1: Black and purple are hard to distinguish. Also, and importantly, it's not clear what the panels in the two rightmost columns are plotting. It seems that these are the first and second eigenfunctions (or principal components as called here) of G ; however, this does not correspond to what is in Table 1 which shows such eigenfunctions. Also, it would be convenient if figures and tables all presented the three crosses in the same order.

We are sorry that our graphical representation was misleading here. To answer to Referee's first comment, the two rightmost columns correspond to the results from the IDM. As such, they represent results only from the phenotypic covariance matrix and not G , which explains

why the results are different from those reported in Table 1 (where the eigen-analysis of \mathbf{G} is shown). We added in the Fig. S1 caption that the IDM is applied to *phenotypic* growth data to avoid confusion.

We thank the Referee for pointing out the issue of population ordering in Figures and Tables. We now homogenized tables and figures so that they report the same order of population (i.e., HELxBYN \rightarrow HELxPYÖ \rightarrow HELxRYT).

As for the colors of the figures, we have changed the purple and black to blue and red (respectively) and also lightened the background of the panels for better clarity.

#18. L 242: It seems that “stated” in L 242 should be “modified” or something like that, as the model in eq. 8 is different to that in eq. 7.

This sentence has been removed from the text and the equation moved to the *Supplementary methods*. Following the Referee’s comment the sentence now reads: “*Consequently, we estimated these allelic effects on the phenotype of individuals using a re-formulation of model (4)*”

#19. L 245: What is z ?

This equation has been moved to the *Supplementary methods*.

z is the genotype coding system matrix associated with the dominance effect γ and coded as stated on L149 in the *Supplementary methods*.

#20. L 270: Here and throughout the term “growth trajectory” is used to mean something that is not clear. The term growth trajectory suggests what is in Fig. S1 panels A, B, C, rather than what is in panels G,J, etc. It should be clarified what is meant by “growth trajectory”, and preferably use more precise terminology, such as eigenfunction or eigenvector.

We thank the Referee for this suggestion. We decided to use the term ‘growth trajectory’ to refer to the results of the IDM analysis (instead of eigenvector, as suggested by the Referee) for two reasons: i) to employ a similar terminology used in the article by Kuparinen & Björklund (2011) where the method is originally described and ii) to avoid possible confusion with our other eigenanalysis of the genetic covariance matrix \mathbf{G} .

However, we understand that this was in itself confusing. To avoid confusion, we now define the term as “eigenvector of the phenotypic variance-covariance matrix” (L250-251). We also replaced the term ‘growth trajectory’ by ‘first eigenvector of \mathbf{G} ’ on L340.

#21. L 284: “size” should be “breeding value of size” or something like that.

We would like to retain the term ‘size’ here as we think it is simpler to describe the genetic correlations between each pair of traits, corresponding to each size at each age.

#22. L 332: I think “would selection act” should be “were selection to act”.

This has been corrected following the Referee's comment.

#23. L 361. Is 66 the right citation? 66 is about mute swans rather than stickleback populations mentioned in the sentence.

We thank the Referee for finding this mistake, reference 66 should not have appeared here. We have now carefully reviewed the citations/references of our manuscript.

#24. L 383: I think "about" should be "of".

This has been corrected.

#25. L 392: "ontogenetic genetic heterogeneity" is quite a mouthful. How about "genetic heterogeneity across ontogeny".

This text has now been removed in our revised Discussion.

#26. Table 1: "loading coefficients" is not very clear. Do you mean "entries"? Also, it would be stricter to speak of eigenvalues, eigenvectors, or eigenfunctions rather than principal components.

We replaced the term "loading coefficients" by the more appropriate "trait loadings" term in the Table title and caption. We have also reworded the title and caption of the figure following the Referee's suggestion, which now reads "[...] *of the two first eigenvectors of the principal component analysis of G* ".

In line with the Referee's comment, we have also reworded the Methods section relating to this analysis and hopefully provide clearer meaning to these terms.

#27. Figure 3: Define P.

Thank you for pointing out this mistake. "P" corresponds to the p value on the logarithmic scale as noted on the y axes of Fig. 3 (now Figure 2) " $-\log_{10}(P)$ " and was missing from the parentheses in the caption of Figure (now Figure 2 in the revised version). This has been corrected.

#28. Figure S4: Transpose the arrangement of the panels to keep consistency with Fig. S3 (rows are genetic variance and columns are heritability). Again, present the crosses in the same order throughout the paper.

This has been corrected following the Referee's suggestion and see also our previous response to comment #17 on the matter.

Referee 2

Main comments: In their paper, Fraimout et al. investigated the genetic architecture of body

size across ontogeny in a stickleback fish. The study uses extensive data from F2-generation inter-population crosses of freshwater and marine *Pungitius pungitius*. They highlight that different QTLs contribute to the heritability of body size in different time-points. However, the heritability of body size remains constant across ontogeny. These results are very interesting in the context of predicting the response to selection in age-structured populations, and more generally, to question the relevance of classical quantitative genetic models, which assume that evolutionary parameters do not change over ontogeny. However, I am not quite convinced by the general presentation of the study, neither by the interpretation of some results. I also have a few comments about the statistical analyses used.

We thank the Referee for their astute comments on our manuscript and we are pleased to read that they found our results interesting. Hopefully the revised manuscript will address the points which left the Referee unconvinced.

#1. My first major comment is about the presentation of the study: the introduction and discussion are both focussed on parallel evolution, that is the repeated occurrence of similar phenotypic or genetic adaptation in indecently evolving populations. I understand that the stickleback has provided a very interesting study system to investigate parallel evolution, however the link between ‘age-dependent genetic architecture’ and ‘nonparallel evolution’ is far from obvious. Assuming a variation in heritability with age, nonparallel evolution would only occur if the independently evolving populations had different age-structures, or if they encountered different selection pressures. Therefore, I think the authors should probably reframe their story, and I would recommend addressing the question of the manuscript in a more general context of linking developmental processes and quantitative genetic theory.

After reading this comment – and others along the same line of thinking from the Editor and the two other Referees – we now see that our previous version of the manuscript lacked a clear focus. Following the Referee’s suggestion (and see also our responses to the Editor and Referee 1 & 3), we have made an effort clarifying the rationale in the Introduction and Discussion sections.

We understand the recommendation of the Referee that our manuscript could be reframed in a more general context of developmental quantitative genetics. Nonetheless, we strongly believe that our results have implications for the study of parallel evolution and we are now hopefully making this point clearer, in the revised Introduction and Discussion.

Specifically, we start by defining the predictability of evolution as the probability of parallel evolution to occur. In other words, evolution would be predictable if the same genomic regions (SNPs or QTL) were being repeatedly used in independent adaptive events. The probability of such repeated use of genomic regions increases with the homogeneity of the pool of genetic variation available for selection to act on in the ancestral population(s). Conversely, heterogeneous standing genetic variation reduces the likelihood of parallel evolution (Fang *et al.* 2021).

Here, we propose that age-specific genetic architecture would constitute such a heterogeneous pool of genetic variation and would consequently decrease the probability of parallelism.

We also acknowledge that the latter statement implies that age-specific genetic architecture would occur in conjunction with temporal selection acting on age-structured populations. The Referee rightfully mentions that “*nonparallel evolution would only occur if the independently evolving populations had different age-structures, or if they encountered different selection pressures*”. We entirely agree with this statement and we believe this scenario is very likely to occur in stickleback fish (see the references by DeFaveri *et al.* (2014), Paccard *et al.* (2018) and Wasserman *et al.* (2021) added on this matter L391-393) and also in other wild species.

Following this scenario, age-specific genetic architecture – as demonstrated by our results – would affect the predictability of evolution as we propose it in the manuscript.

References

- Fang, B., Kemppainen, P., Momigliano, P. & Merilä, J. 2021. Population structure limits parallel evolution in sticklebacks. *Mol. Biol. Evol.* **38**, 4205-4221. (doi: 10.1093/molbev/msab144)
- DeFaveri, J., Shikano, T., & Merilä, J. 2014. Geographic variation in age structure and longevity in the nine-spined stickleback (*Pungitius pungitius*). *PLoS One*, **9**, e102660. (doi: 10.1371/journal.pone.0102660)
- Paccard, A., Wasserman, B. A., Hanson, D., Astorg, L., Durston, D., Kurland, S., ... & Barrett, R. D. 2018. Adaptation in temporally variable environments: Stickleback armor in periodically breaching bar- built estuaries. *J. Evol. Biol.* **31**, 735-752. (doi: 10.5061/dryad.7h4s265)
- Wasserman, B. A., Reid, K., Arredondo, O. M., Osterback, A. M. K., Kern, C. H., Kiernan, J. D., & Palkovacs, E. P. 2021. Predator life history and prey ontogeny limit natural selection on the major armour gene, Eda, in threespine stickleback. *Ecol. Freshw. Fish.* (doi: 10.1111/eff.12630)

#2. The study also needs to be more rooted in the existing literature about the ontogeny of genetic effects (a few references provided below). For instance, the authors never compared their results with previous studies reporting estimates of additive genetic variance or heritability at different ages. We need such a comparison to critically discuss the results found here.

Ontogeny of genetic effects, just a few examples: Cheverud *et al.* 1983 *Evolution* 37, 895-905; Gauzere *et al.* 2020 *Evolution*, doi: 10.1111/evo.14000 ; Lindholm *et al.* 2006 *Biology Letters*, doi:10.1098/rsbl.2006.0546 ; Wilson & Réale 2006 *American Naturalist*, doi: 10.1086/498138.

We acknowledge that our manuscript lacked a number of important references regarding the evolution of genetic variation during ontogeny in other study systems and agree with the Referee that we should have discussed our results more with the existing literature. We would like to provide two main response to the Referee’s comment:

First, we now have incorporated more references suggested by the Referee in our text. In the Introduction, we now acknowledge that similar work has been conducted in the past to

estimate genetic variances (additive, maternal etc.) with age in a number of wild and domesticated systems.

Second, while the Referee is right that our results can be discussed in the light of these studies, we respectfully argue that such discussion is slightly outside of the scope of the current study. For instance, the changes (or lack thereof) in additive genetic variance and heritability we observe with age of the sticklebacks could be discussed against theories such as that of Medawar (1952) and Williams (1957) (Mutation accumulation and antagonistic pleiotropy, respectively) and with the large body of literature on the quantitative genetics of senescence. However, rather than focusing on the changes in genetic variance and heritability with age, we want to show that body size is heritable throughout ontogeny in spite of being underlined by different QTL and can therefore respond to selection at different ages.

We have added text in our Discussion section in line with the response above (see L370-376).

We hope that these changes will provide a better framing to our study and root our results in the literature.

References

Medawar, P. B. *An Unsolved Problem of Biology: An Inaugural Lecture Delivered at University College, London, 6 December, 1951* (H. K. Lewis, 1952).

Williams, G.C. 1957 Pleiotropy, natural selection, and the evolution of senescence. *Evolution*, 398-411. (doi:10.2307/2406060)

#3. My second major comment is about the interpretations of the association mapping. From these analyses, the authors claim that ‘different QTL control body size in different ages, and there were not global QTL which could influence growth in two or three crosses’ (l. 374-376). Knowing the statistical biases of mapping, I am surprised that the authors are not discussing more critically their results. They reported a total of 11 unique QTLs significantly associated with body size variation (l. 303-304). However, they never report the estimated effect sizes, neither the total amount of genetic variation explained by these significant QTLs.

The Referee raises an important point here and we have now addressed this comment cautiously in our revised version. First, we now report all effect sizes and proportion of variance explained (PVE) by each significant QTL in our new Table S3, along with graphical representations of the QTL’s effect sizes in the *Supplementary material* (Fig. S6 – S8).

Second, we now address the issue of statistical power up-front in our revised Discussion (see L325-331 and see also our response to Referee 3 comment #2) and acknowledge that our mapping analyses may suffer from low power.

More importantly, we also discuss how the other results of the study provide support for age-specific genetic architecture. Specifically, the absence of global QTL and the non-significant genetic correlations between size at different ages, both suggest that the genetic architecture of size is not homogeneous across ontogeny.

We hope that this changes will address the Referee’s concern satisfactorily.

#4. I can imagine that only a small fraction of VA is explained by the detected QTLs. If so, the results most likely support a polygenic architecture of body size, and the loci underlying trait variation should be very difficult to detect significantly given their low effect sizes.

In agreement with the Referee's comment, most significant QTL mapped in our study crosses explain small to moderate amounts of variance in body size at each age (see new Table S3 in the revised manuscript).

Therefore, we agree with the Referee that these results support a polygenic architecture of body size (which we now acknowledge following the Referee's comment; see L326-328) .

Nonetheless (and see the revised text L333-350; our above response to comment #3 and response to Referee 3's comment #2) the age-specific genetic architecture we observe is also corroborated by several other results in our study, and not restricted to the QTL mapping analysis: The absence of a global QTL for the growth rate parameter k (keeping in mind the potential statistical power issue we may face; now discussed L325-331) and the lack of significant genetic correlations between ontogenic end points both support a heterogeneous genetic basis for body size across ontogeny.

#5. For polygenic traits, we know that the results of independent mapping studies can show strong divergence. In such context, the quite consistent effect of Chr. 9 at different ontogenetic stages in the HEL x BYN and HEL x PYÖ crosses could support the evidence of one global QTL influencing body size (yet explaining a small fraction of VA).

We agree with the Referee that the repeated effect of the QTL on Chr. 9 in both crosses could support the evidence for a global QTL controlling body size variation. However, we would like to respectfully argue this point with two results which, in our opinion, do not support the presence of a global QTL: i) we did not find any significant QTL underlying the expression of the growth rate parameter k (a proxy for general growth) and ii) the effect sizes of these QTL in the two crosses are in fact opposite (see Table S3 and Fig S6 – S8).

Following the Referee's comment, we have added text in line with this response to the Discussion (see L335-338).

#6. Concerning the statistical analyses, I understand the relevance of treating growth measurements either as a 'character state trait' or a 'function-valued trait' (l. 161-163). However, I think the authors need to provide more information about this new method implemented in the dynGP package, and it should be stated clearly that this method estimates a 'SNP-heritability' (as I understand from reading Arjas et al. Bioinformatics 2020). SNP-heritability and the trait heritability parameter have different definitions (de los Campos, Sorensen, Gianola Plos Genetics 2015), and therefore the authors probably need to compare these two estimates.

Thank you for this important remark. The Referee is right in mentioning that the original paper by Arjas *et al.* (2020) clearly states that their method estimates 'SNP-heritability', as mentioned on p. 3795 of their article: "*We note, that in case heritability is estimated from*

genomic data, it is called SNP-heritability, and we will from now on refer by heritability specifically to narrow-sense SNP-heritability.”

There is, in our opinion, an important point to note here: our animal model approach uses genomic data (the GRM) in a linear mixed model (LMM) framework to estimate heritability. Similarly, *dynGP* is based on the LMM framework and uses also the GRM as is stated p. 3797 in Arjas *et al.* (2020): “*All models in this study are extensions of the basic LMM [...]*”. Following this logic, we should also coin our heritability estimates from GRM-based animal models as ‘SNP-heritability’.

However, the article by de los Campos *et al.* (2015) rightfully mentioned by the Referee emphasizes the importance of the statistical model behind heritability estimation in describing the parameters of interest. Here, both approaches used (*MCMCglmm* and *dynGP*) use marker information to describe the relatedness among individuals, but they do not constitute a direct multi-locus association (MLA; *sensu* Arjas *et al.* 2020) and the markers are thus used as surrogates for the causal variant underlying variation in the trait. As such, both models relate to the classical quantitative genetics theory and we believe that for both approaches, the estimated heritability should be coined ‘heritability’ (at least, in the present study).

All this said, we have addressed the Referee’s comment in two ways: i) for the sake of clarity we have removed the term ‘SNP-heritability’ from the main text and thus consider all our estimates to be equivalent measures of heritability (both based on *MCMCglmm* and *dynGP*); ii) in the new *Supplementary methods* section we now explain in detail the *dynGP* method and provide more technical information about this analysis.

#7. I also wonder why the authors have not used the antedependence model implemented in *MCMCglmm* to analyse their data (see Thomson *et al.* Evolution 2017 for an example). From what I understand, this type of model is the most appropriate to deal with age-specific random effects, while accounting for the non-independence of the effects. I think the authors need to better justify their approaches.

We thank the Referee for pointing out this alternative analytical approach for our data. Indeed, the antedependence model in *MCMCglmm* seems particularly suitable in our case and we followed the Referee’s suggestion to try and implement it in our study.

First, we have attempted to fit an antedependence structure in *MCMCglmm* (following recommendations by Jarrod Hadfield; pers. comm.) while retaining the GRM in the random effect term. Unfortunately, the computing time makes it virtually impossible to run the model for a sufficient number of iterations (in fact, figuring out what the sufficient numbers of iteration would be is in itself unattainable). To give precise numbers: running the model on one of our cluster’s supercomputers (Xenon Gold 6230; 2x20 cores @ 2,1 GHz), we were only able to obtain 34,000 MCMC iterations after 72 hours of computing. Thus, obtaining reasonable posterior samples (with MCMC iterations above the order of 100K) seems rather challenging.

We thus fitted the same model structure this time using only the pedigree information in lieu of the GRM. This model type had a much more efficient computing time (ca. $2e10^6$ iterations

in ~20 hours). However, we were unable to obtain meaningful estimates of variance components and heritability (see Figure below). Although the antedependence model provided estimates supporting our other models (uni- and multi-variate *MCMCglmm*; *dynGP*) for an increase in V_A with age, these estimates seem particularly inflated (possibly due to a strong underestimation of residual variance). Consequently, heritability was largely overestimated and was not consistent with the values obtained from animal models and PVE from QTL analyses.

Results from the antedependence model structure in the HELxPYÖ cross. The median of the posterior distribution of all variance components (colors) are shown for each age-specific size based on a model using the antedependence structure in *MCMCglmm* and the Pedigree Relationship Matrix (PRM). Vertical bars represent the 95% HPD intervals.

This result suggests that the information contained in the pedigree for single families is insufficient to partition the phenotypic variance into meaningful components (and please see our response to comment #12 below regarding this matter).

Consequently, we respectfully ask that we should retain the current modeling approach in our manuscript.

Detailed comments:

#8. l. 77-79: I think this is true only if the different populations have different age-structures.

We agree with the Referee on this point. In the specific case of the nine-spined sticklebacks, among-population variation in age-structure is in fact expected, as shown by a previous study from our research group (de Faveri *et al.* 2014).

Furthermore, temporal variation in selection has been demonstrated in several other systems (reviewed in Siepielski *et al.* 2009), including sticklebacks (Paccard *et al.* 2018, Wasserman *et al.* 2021), and we thus think that the conceptual idea of age-specific selection acting on age-specific genetic architectures that we bring up in the Introduction should hold true.

Following the Referee's helpful comment, we added text in our Introduction in line with the above response and added the references above in our Discussion (and see also our response to comment #1).

References

DeFaveri, J., Shikano, T., & Merilä, J. 2014. Geographic variation in age structure and longevity in the nine-spined stickleback (*Pungitius pungitius*). *PLoS One*, **9**, e102660. (doi: 10.1371/journal.pone.0102660)

Siepielski, A.M., Morrissey, M.B., Buoro, M., Carlson, S.M., Caruso, C.M., Clegg, S.M., Coulson, T., DiBattista, J., Gotanda, K.M. & Francis, C.D. 2017 Precipitation drives global variation in natural selection. *Science* **355**, 959-962. (doi:10.1126/science.aag2773)

Paccard, A., Wasserman, B. A., Hanson, D., Astorg, L., Durston, D., Kurland, S., ... & Barrett, R. D. 2018. Adaptation in temporally variable environments: Stickleback armor in periodically breaching bar- built estuaries. *J. Evol. Biol.* **31**, 735-752. (doi: 10.5061/dryad.7h4s265)

Wasserman, B. A., Reid, K., Arredondo, O. M., Osterback, A. M. K., Kern, C. H., Kiernan, J. D., & Palkovacs, E. P. 2021. Predator life history and prey ontogeny limit natural selection on the major armour gene, *Eda*, in threespine stickleback. *Ecol. Freshw. Fish.* (doi: 10.1111/eff.12630)

#9. l. 125-128: Why not having use the same protocol for all crosses?

The crosses used in the current study were used in previous studies by our research group on the genetic architecture of different traits in *P. pungitius* and were therefore produced with different protocols. In the current study we re-analyze these crosses.

We also note that due to page limit restrictions, part of this text was moved to the *Supplementary methods*.

#10. l. 191-194: Probably this two-step analysis is not necessary if the authors model an

antependence structure directly in their MCMCglmm model (see Thomson et al. Evolution 2017 Supporting Information).

Please see the response to comment #7 above on this matter.

#11. 1. 195-204: Please provide more information about this new method, especially because it does not estimate the same heritability parameter than the MCMCglmm models. Then, how the analyses considering growth measurement as a ‘character state trait’ or a ‘function-valued trait’ can be compared?

We now provide a more detailed description of the *dynGP* method in the new *Supplementary methods* section and have also decided for the sake of clarity to refer to all heritability estimates in our manuscript as ‘heritability’ (see our response to comment #6).

#12. 1. 223-226: I understand that multivariate animal models using GRM are computationally intensive. Using animal model with pedigree data is much more efficient. Maybe using a pedigree data would then allow to use all growth measurements. Have the authors tried to compare models using GRM and pedigree data?

This is a very good point and we have indeed compared both types of relatedness matrices. In fact, we currently have a manuscript under review on this topic (Framout *et al.* 2021a; *Dissecting the genetic architecture of quantitative traits using genome-wide identity-by-descent sharing among full-sibs*. bioRxiv. doi:10.1101/2021.03.01.432833) which shows that for these specific crosses, the pedigree-based animal models do not allow for the estimation of meaningful quantitative genetic parameters. See below an example from this study illustrating this point with focus on three morphological traits including body size at adult age as used in our current manuscript.

Comparison of variance and heritability estimates using different genetic relationship matrices. The modes of the posterior distribution of all variance components are shown for each trait based on a model either based on the GRM (dark blue) or the Pedigree Relationship Matrix (PRM; light blue). Black bars represent the 95% HPD intervals.

Therefore, we are confident that GRM-based models are the best approach here.

#13. 1. 279-281: SNP-heritability are supposed to be reported in Fig. 1, Fig S2-S4. However, the captions of these figures indicate that only the *MCMCglmm* results are reported. Please, indicate more clearly where these estimates can be found.

We now realize that our wording was misleading in this section of the manuscript and elsewhere (see our response above on this matter). We indeed meant that these figures report the results from *MCMCglmm* models and therefore not the SNP-heritability. We have corrected this wording which now simply reads “Heritability”.

#14. 1. 292-294: how this result relates to other studies analysing the dimensionality of \mathbf{G} matrices?

In our case, as most genetic variance is explained by \mathbf{g}_{max} , \mathbf{G} matrices have a high eccentricity (*sensu*, Guillaume & Whitlock (2007). Effects of migration on the genetic covariance matrix. *Evolution*, 61(10), 2398-2409). As to the comparison with other studies of the dimensionality of \mathbf{G} , because our inter-population crosses are not representative of natural populations *per se*, we prefer to remain cautious in comparing our results to other systems (obtained from wild populations) and focus on the result which pertains most to our study:

there is ample genetic variance underlying body size throughout ontogeny, thus body size can theoretically respond to selection at all stages of development.

We also note that with all requests made by the Referees, the length of the manuscript is already close to exceeding page limits for Proceedings B.

#15. 1. 303: The authors should also report the total amount of genetic variance in body size explained by these 11 QTLs.

This is now reported in our new Table S3.

#16. 1. 381-382: the authors need to better explain why maternal effects are not expected to be important here, as we know that juvenile traits are strongly affected by maternal effects. How the experimental design / models manage to dissociate maternal genetic effects and direct genetic effects?

Our wording here was misleading, as evidenced by the Referee's comment and we have now rephrased this sentence. Environmental and maternal effects should not be of great concern in our estimations but we now clearly explain why. First, the estimation of genetic variance from within-family variation using the actual IBD relationships between full-sibs should not be affected by common environmental effects (Visscher *et al.* 2006; Ødergård & Meuwissen 2012). Given that all individuals were raised in a common-garden, environmental variance should also be minimal in our data.

Second, regarding the possible maternal effects, although our crossing design does not directly allow for the partitioning of maternal variance, previous work by our research group showed that such effects are expected to be minimal for body size in both marine and freshwater populations of *P. pungitius* (Ab Ghani *et al.* 2012).

We have now reworded this part of the text and added the above references to better explain our point (L382-384).

References

Visscher, P.M., Medland, S.E., Ferreira, M.A., Morley, K.I., Zhu, G., Cornes, B.K., Montgomery, G.W. & Martin, N.G. 2006 Assumption-free estimation of heritability from genome-wide identity-by-descent sharing between full siblings. *PLoS Genet.* **2**, e41. (doi:10.1371/journal.pgen.0020041)

Ødergård, J., & Meuwissen, T. H. 2012. Estimation of heritability from limited family data using genome-wide identity-by-descent sharing. *Genet. Sel. Evol.* **44**, 1-10. (doi: 10.1186/1297-9686-44-16)

Ab Ghani, N. I., Herczeg, G., & Merilä, J. 2012. Body size divergence in nine-spined sticklebacks: disentangling additive genetic and maternal effects. *Biol. J. Linn. Soc.* **107**, 521-528. (doi: 10.1111/j.1095-8312.2012.01956.x)

#17. Figure 1: representing CVP (or CVR) would be relevant to fully interpret the results. I think the authors also need to clearly explain in their main text that the variation in CVA is not associate with a change in h^2 because of an increase in CVP.

Thank you for this helpful remark. We have now re-designed Figure 1 and included both CV_P and CV_R . We also now state in the main text that both CV_A and CV_P increase with age and consequently that heritability remains relatively constant.

Note that because of page limits we have moved Fig. 1 to the *Supplementary material* section (now Fig. S2).

Referee 3

This manuscript deals with an emerging principle that is important for a fundamental understanding of phenotypic evolution: that similar phenotypic end-points can be reached with diverse genetic mechanisms. The results of this study support this notion with classical quantitative genetic and QTL analyses in an ontogenetic trait: growth of body size in nine-spined sticklebacks. The study is impressive in the range of different techniques and modelling approaches that it combines. But this is also one of its weaknesses: many of the methods are barely explained and the results obtained barely discussed (for an example, take the Infinite Dimensional Model). As such I found it quite difficult to understand the paper.

We thank the Referee for the helpful and relevant comments received on our manuscript. We apologise that the manuscript was in several instances difficult to understand. Hopefully this revised version satisfactorily addresses all points raised by the Referee.

To my mind, three changes would make this a more accessible and convincing contribution to the field:

#1 1) The methods need to be explained in a lot more detail, where necessary in the Supplementary Material.

We now provide a *Supplementary methods* section which details the approaches and analyses used in our manuscript. As responded to the Editor and other Referees we also have made a particular effort in providing both accessible methods explanation and justification in the main text, and detailed technical explanation in the *Supplementary methods*.

#2 2) An alternative explanation for the results needs to be addressed head on: That the observed results are largely a consequence of the low statistical power that is evident in the wide confidence intervals of the estimates of heritability and CV_A . This makes the detection of any pattern difficult and it would also explain why different QTL pop up at different ages and in different crosses. With limited statistical power and many small-effect loci affecting a trait like body size, it is fairly random which loci pass the significance threshold, leading to the apparent differences in genetic architecture across ages. The authors touch on this in the Discussion, but to my mind the manuscript would benefit from a more in-depth discussion of this possibility.

The Referee raises an important point here (as also pointed out by Referee 2 and see our response to their comment #3 and #4).

We acknowledge that QTL mapping will always be restricted by sample size and given the polygenic nature of body size (and see the added Table S3 of all QTL effect sizes and PVE), the QTL mapping analysis presented here may suffer from a statistical power issue. Following the Referee's comment, we now address this possibility head on and we have added text to our revised Discussion (L325-331) along this line of thinking.

Despite the possibility of low statistical power, we also discuss the fact that other results from the study support the presence of age-specific genetic architecture. To summarize (but see the added text L333-350): (i) we did not detect any global QTL explaining variation across the whole ontogeny in any of the three crosses. This suggests no common genetic basis for growth. (ii) Genetic correlations between age-specific body sizes were consistently decaying with time and were often found to be non-significant (in spite of the large confidence intervals around posterior estimates; see also below), and thus indicating different genetic bases for body size variation between different ages. (iii) Our QTL-mapping approach associated with Linkage Disequilibrium network (LDn) clustering should be particularly efficient in addressing issues of false-positive/missing QTL (and see Li *et al.* 2018).

Therefore, we are confident that our QTL-mapping should capture truly different QTL between crosses and ages.

We also acknowledge that our estimates of variance components present wide confidence intervals and that patterns of variation across ontogeny are more difficult to observe, particularly for heritability estimates. In line with this and following the Referee's comment we have toned down the focus on constancy of heritability throughout the manuscript and in the title.

Nonetheless, the relatively low statistical power of our animal models does not discard three main findings of our analyses: i) heritable variation and additive variance are underlying body size across ontogeny (i.e., the confidence intervals of these two parameters are substantially above zero in most estimations); ii) V_A in late age is significantly increased compared with early age in two of our three crosses (new Fig. S2) as evidenced by the non-overlapping confidence intervals between age 1 (Week 02/Week 04) and age 7/9 (Week 26/Week 34) in HEL x PYÖ and HEL x RYT, respectively and iii) genetic correlations are decaying with age and are not significantly different from zero between early and late life body size in two out of three crosses.

In line with this response we have added text in our revised Discussion (L370-376).

#3 3) There is a large body of literature on the quantitative genetics (incl. QTL) of growth in livestock. This manuscript makes few references to that literature. I think this manuscript would gain substantially if both methods and results were put into the context of what is known from livestock.

In line with the Referee's comment and with Referee 2' similar comment, we now provide a wider set of reference against which we compare and discuss our findings. These also include literature from the livestock research as suggested by the Referee. However, we need to limit the number of references as the asked revisions are pushing the manuscript beyond page length limits.

DETAILED POINTS:

#4 Introduction: The Introduction has a conceptual 'fault line' that I found distracting. The first several paragraphs focus on the predictability of phenotypic evolution. Starting on line 76, there is a sudden and unexplained switch to considering the similarity of genetic architectures underlying the same phenotype. These two aspects are also intermingled on lines 91-96, where lines 91-93 seem to focus on the predictability of phenotypic evolution, while the following lines look at genetic architecture. To me phenotypic evolution and its genetic underpinnings are two very different perspectives that should not be mixed casually. I would suggest to focus from the outset on the similarity of the genetic architecture underlying the same phenotype, just as is done in the opening of the Discussion on lines 322-324.

We thank the Referee for these suggestions. In line with this, and also following both other Referees' and the Editor's comments, we have substantially re-written this section of the Introduction.

Specifically, we now emphasize the link between the two notions at play in our manuscript: age-specific genetic architecture and the likelihood of parallel evolution.

As we state it, the link between these two notions is rather straightforward: an increase in heterogeneity of the genetic architectures of similar phenotypes will decrease the probability of parallel evolution (and with it, the predictability of evolution). In this context, age-specific genetic architectures would constitute a more heterogeneous pool of genetic variation for selection acting on age-structured population. Therefore, age-specific genetic architecture could decrease the probability of parallel evolution, and also, the predictability of evolution.

We hope that the added text in the revised Introduction (particularly L76-L82) and Discussion (L388-396) will clarify our points.

#5 L49-55: Presumably for brevity's sake, the authors use an all or none language here: is evolution predictable and will species adapt to changing environmental conditions? Although it may increase the word count, I think a more quantitative wording would be appropriate for the topic in general and this particular contribution: To what extent is evolution predictable and to what degree and how fast will species adapt to changes?

We have expanded on the idea of the predictability of evolution in our revised Introduction (in line with the Referee's question here) and Discussion.

To summarize (but please see the revised text L64-67; L76-82; L388-396 and the response above to comment #4 and to Referee 2's comment #1, #2 and our response to the Editor) we argue that age-specific genetic architecture should decrease the predictability of evolution by decreasing the probability of parallel evolution. Age-specific genetic architecture for the same quantitative trait provides a heterogeneous pool of genetic variation for selection to act on and therefore, decreases the probability that similar genes would underlie the same adaptive phenotype.

That being said, although it was not our initial goal to aim for lower word count in our Introduction, the substantial revisions we made to our manuscript now constrains us to be more concise. We therefore respectfully ask that this added text on the predictability of evolution would suffice to address the Referee comment.

#6 L59: See the previous comment. It would be nice to view the degree to which evolution is predictable quantitatively rather than qualitatively.

See our previous response. We hope that our revised version provides better explanation for the way our results articulate with the predictability of evolution.

L75: Should read: '... paid to...'

This has been corrected.

L110: In what sense were these crosses independent given that in all three crosses one of the parental populations was the Helsinki population?

Our wording was misleading here as we referred to the independence based on the pond paternal origins (which are arguably, independent due to the high genetic divergence among them). We therefore removed the term 'independent' from the text.

L134: 20'000 loci is a sizeable number of SNPs but by today's standards it is not a large number.

We have removed the word "large" from this sentence. We also note that the *SNP genotyping* section has been moved to the *Supplementary methods* due to page length restrictions.

L155: Please explain the purpose of the Infinite Dimensional Model.

We have reworded this sentence to better justify the use of the IDM and we now also provide detailed explanation of this approach in the *Supplementary methods*.

L167-170: Please make it clear that here you are considering body size a character state trait and that you ran a separate model for each age. Also, please detail what beta and gamma were made up of and what distributional assumptions you made.

We now clearly state that we ran separate univariate models for each size at each age and we further detailed the definition of beta and gamma, following the Referee's comment. The sentence now reads: "[...] where y is the vector of phenotypic values for age-specific body size, β is the vector of fixed effect, γ is the vector of random effects, ϵ is the vector of residual errors and X and Z are the design matrices relating to the fixed and random effects, and corresponding to the individual values of body size at one age and the matrix of relatedness coefficients, respectively."

We now also specify the distribution assumed by our models.

L175-176: This explanation of the GRM was impenetrable for me. Please explain at a level of an average reader of the Proceedings.

We now include a more accessible description of the GRM in the main text and detailed information regarding its construction in the *Supplementary methods*.

L177-178: Please justify these filtering choices.

We have now moved this section of the text to the *Supplementary methods*. The filtering values were set using the *raw.data* function of the *snpReady* package, which we now state in the corresponding *Supplementary methods*.

L180: Please specify and justify exactly which priors you used. *mcmcglmm* offers a variety of priors that are non-informative to varying degrees.

We have reformulated this sentence and added this information to the text. We used flat priors in *MCMCglmm* by setting the degree of belief parameter *nu* to 0.

L190: Is BSA the same as y in eq. 2?

Yes, it is the body size measurements at a given age.

L191-194: This was impenetrable for me. Please keep an average reader of such an article in mind.

We apologize for the confusing writing. The smoothing step allows to compare the results obtained from the ‘character-state’ approach (i.e., the separate univariate models) to the ones obtained from the ‘function-valued’ approach (i.e., the *dynGP* analysis). The smoothing step is applied to the set of character-state observations (the set of MCMC posterior for each variance component) to derive the pattern of change of these observations with age.

We have now re-worded this section of the text to better justify this analytical step.

L201: Why use a different way of estimating the GRM in the two different analyses? This makes it harder to compare the results of the two analyses.

This is simply a formatting issue: *snpReady* requires that the SNP data is coded as 0, 1 and 2 for the genotypes AA, AB and BB, respectively, whereas *dynGP* requires the SNP data to be coded as -1, 0 and 1 for the same genotypes. The *rrBLUP* packages allows for the construction of the GRM using the latter -1, 0, 1 format and we chose this package to build the GRM used for the *dynGP* analysis, as well as to follow the same approach than in the original publication.

As such, both GRM are built from the exact same SNP data but are simply coded differently and are providing comparable results. See below the raw graphical output results obtained for the HELxPYÖ with the same GRM as in equation (2) and which are identical to the results from Figure S2:

Results from the *dynGP* analysis based on the GRM constructed from *snpReady* in the HELxPYÖ cross. The mean residual variance (“meanse”), additive genetic variance (“meansg”) and mean heritability (“meanher”) are shown throughout development (“age”). The mean value of the posterior distribution (blue solid lines) is shown along with the 95% confidence intervals (dashed red lines).

L203: This wording suggests that a regression with only an intercept and a term for sex was used, and no slope. Is this correct?

Yes, the residuals were obtained using the following command in R:

```
res = lm(body.size ~ sex)$residuals
```

L208-211: The nature of growth trajectories introduces an autoregressive covariance structure among the epsilons in this formulation of the model. Was an autoregressive covariance structure used for the permanent environment effect?

This is a very good point raised by the Referee and in line with Referee 2’s comment #7. Given the nature of our growth data, an alternative modelling approach would have been to fit a first order autoregressive covariance structure for either the permanent environmental effect or the residuals. As noted by Referee 2, we could have used the antedependence model structure implemented in the *MCMCglmm* package for this purpose. Unfortunately, we encountered 2 main issues which prevented us from using such approach (and please see our response to Referee 2’s comment #7 on this matter): i) using the GRM to summarize the among-individuals’ genetic relatedness provides us with more power to estimate genetic variance components (and see Fraimout *et al.* 2021) but also comes at the cost of computation time. We were thus not able to obtain sufficient MCMC iterations when running this type of model; ii) using pedigree information for this type of cross and running the antedependence model did not yield meaningful results, particularly with regards to residual variance and heritability.

Nonetheless, as all statistical modeling approaches used (i.e. uni- and multivariate MCMCglms, and *dynGP*) converge to similar results, we are confident that the results we report should hold.

L213: Please provide a reference for eq. 6.

We added the following reference to equation 6: de Villemereuil, P. (2018). Quantitative genetic methods depending on the nature of the phenotypic trait. *Annals of the New York Academy of Sciences*, 1422(1), 29-47. This also refers to the implementation of quantitative genetics analyses in R by Pierre de Villemereuil (https://devillemereuil.legitux.org/wp-content/uploads/2021/09/tuto_en.pdf)

L215-223: This section was impenetrable for me. Please explain in more detail what this analysis captures.

We have now reformulated the whole section on the analysis performed on the **G**-matrix. We hope this revised text is clearer.

Also, please clarify how the analysis here is related to the ones in the references (57,58). From my reading, references 57 and 58 use PCA to reduce the dimensionality of marker data, while here PCA is applied to the G matrix estimated from the multivariate animal models. But perhaps I misunderstand?

Unfortunately, there was a mistake in the citations on our part which led to this misunderstanding. We apologize for the inconvenience and have now fixed the citations and references (see also the responses below on this matter).

L241: Is reference 62 (Westfall, P.H. & Young, S.S. 1993) really appropriate here?

The references have been fixed.

L251: The reference to LEP-MAP3 should be 60, and the reference for LDna should be 61. It seems that something is broken with the reference numbering and I will refrain from further commenting on mismatches.

We thank the Referee for pointing out the reference issue. Indeed, something went wrong with the reference number which we have now been fixed.

eq. 7 and 8: Shouldn't there be a sum across loci?

As we used a single locus-based method for QTL mapping, we are analyzing one locus at a time which explains why there is no sum across loci in this equation.

Also we note that due to length restrictions, these equations were moved to the *Supplementary methods*.

L269-273: I found this hard to follow. Please explain in more detail.

We now provide a detailed description of the IDM analysis in the *Supplementary methods* which will hopefully make this section clearer to the Referee. We have also reformulated the justification for the IDM in the Methods section.

L279: I did not see the SNP-heritability results in Figure 1.

We decided to remove the term SNP-heritability from our manuscript for the sake of clarity. Please also see the response to Referee 2's comment #6 on this matter.

L279-281: I found this wording confusing. At least qualitatively both the function-valued trait and the character state trait approach showed an increase of (C)V_A with age, but heritability to remain unchanged. That last conclusion, however, is very uncertain given the very large uncertainties in estimates of h².

The Referee is right in the interpretation of this result. Consequently, (and see our response below) we have toned down parts of the text focusing on the relative constancy of heritability throughout ontogeny. Nevertheless, we still find that the results mentioned in this section (i.e., increase in CV_A in HELxPYÖ and HELxRYT) hold true despite the relatively low statistical power of our animal models.

Figure 2: Please do not omit the estimates of the genetic correlations when they are not significant. Also, does 'multi-trait model' refer to the multivariate model on L208?

We have added the non-significant genetic correlation values back to the Figure 2 (now Figure 1). Yes, the 'multi-trait-model' refers to the multivariate model on L195. We have now clarified this in the figure caption.

L292-293: Is this not simply the definition of the first principle component of a PCA? In other words, the fact that the first principle component explains the most variance is simply a consequence of the definition of a first principle component and not a biological result, no?

We understand how our wording could lead to a misunderstanding here. The Referee is right in saying that by definition a first principal component explains most of the variation present in the data (i.e., PC1 is always > PCn). However here we simply note that a large portion (> 90%) of the genetic variance is explained by the first PC. Another situation could have been that PC1 indeed explains most of the variance in the data (by definition) but to a lesser extent, and that other PCs could explain substantial amount of variance. This would have been the case for example with a non-eccentric **G** matrix (and please see also our response to Referee 2's comment #14 on the matter of matrix eccentricity).

L303-308: Perhaps this textual repeat of the results in Figure 3 are not necessary?

Thank you for this suggestion.

Indeed, and due to the page restrictions of the journal we have decided to remove this textual repeat of the Figure (now Figure 2).

L313: Should this be 'stages' instead of 'changes'?

This has now been corrected following the Referee's comment.

L318-319: Could this be a result of low statistical power?

This is an interesting point. Although we have reasons to believe that there is little contribution of dominance genetic variance for such traits in this cross (see our other study Fraimout *et al.* 2021; *Dissecting the genetic architecture of quantitative traits using genome-wide identity-by-descent sharing among full-sibs*. bioRxiv. doi:10.1101/2021.03.01.432833) it is possible that statistical power – or the type of crossing used – does not allow us to estimate dominance effects accurately.

L325: I do not understand the focus on heritability here, given that levels of additive variation seem a more relevant measure here (see e.g., Houle 1992, Genetics). And levels of additive variation tell a different story (Figure 1): additive variation increases markedly with age in all three crosses. The reason that heritability stays the same is that the phenotypic variation also increased, which makes sense given that growth often magnifies differences over time. Incidentally, I missed a discussion of the differences between the age-related patterns of (C)V_A and heritability in the Discussion.

In line with the Referee's comment, we have now removed the focus on heritability in this part of the Discussion to avoid confusion, and overall toned down the focus on the constancy in heritability throughout the manuscript and in the title.

Regarding the discussion on the differences between age-related patterns of variance components (and please see our response to Referee 2's comment #2): we acknowledge that our results could be discussed in light of the large body of literature on the quantitative genetics of ageing. However, instead of focusing on the age-specific changes in genetic variance and heritability, we focus on the fact that there is heritable variation underlining body size throughout ontogeny and therefore that this trait can respond to selection at different ages. Furthermore, because of the wide confidence intervals around our estimates pointed out by the Referees, it is perhaps best to focus on the result that pertains the most to our study.

We have added text in our Discussion section in line with the response above (see L370-376).

L337-340: The males were sampled from three different ponds, some of them quite far apart suggesting marked genetic structuring among the ponds. Would it not be a more likely explanation that the pond populations provided a more genetically diverse background, leading to the majority of the additive genetic variance to come from the segregation of pond alleles?

We agree with the Referee and have added a sentence in this part of the text (now L319-320) in line with this comment.

L340-341: This argument (that pond alleles contribute to size more strongly than marine alleles) implies dominance effects, but dominance effects were rare. Does this not suggest that dominance of pond alleles is an unlikely explanation, just as is outlined below on L342-343.

Indeed, we agree with the Referee and we have reworded this sentence to make this point clearer.

L385: They are few indisputable facts in science.

We have toned down this sentence.

Appendix B

Associate Editor

Two of the referees who evaluated your original manuscript have now provided comments for the revised version. As you will see, they are very positive but referee 2 makes a very important point that needs to be addressed. You do need to acknowledge the fact that numerous existing studies focused on farm animals have already demonstrated empirically that the genetic architecture of growth is ontogenetically heterogeneous in several species.

Your study would be made even more valuable by discussing the existing reported evidence for ontogenic heterogeneity in genetic architecture of size/growth and contrasting it with your results as it represents a very thorough and rigorous demonstration that this process can be important for local adaptation, which goes well beyond what other studies have done.

We thank the Editor and both Referees for their positive comments on our revised manuscript and we are very pleased to understand that most of our revisions were satisfactory. In the revised version we have addressed the major comment of Referee 2 regarding the existing literature on heterogeneous genetic architecture of growth in two ways: i) we toned down throughout the manuscript any claims of novelty for the finding of age-specific genetic architecture and instead, emphasized that this aspect has seldomly been addressed in the context of local adaptation and parallel evolution and ii) we now discuss our results in light of previous findings from the domestic animals literature and added text in the Discussion including helpful citations from the Referee. Nevertheless, we respectfully ask that we could keep this aspect of the discussion concise – due to page limitations – as the previous round of revisions led us to substantially reduce the length of the text already.

Referee 2

I really appreciate the effort the authors took to improve their presentation. They have addressed my previous comments thoroughly, and it seems they did the same with the comments from other reviewers. Therefore, I have no further comments.

We are very pleased to have addressed all of the Referee's comments, thank you again for your critical and constructive review.

Referee 3

The authors have substantially revised the manuscript. It is now considerably easier to read and understand and I thank the authors for this effort.

Thank you for this positive comment, we are pleased that our revisions were satisfactory to the Referee.

I have two main comments on the revised version of the manuscript.

1) The authors failed to address one important issue in the revision. The manuscript continues to ignore the large body of literature on QTLs of growth in farm animals. Studies of farmed animals (including fish) over the past nearly 20 years have shown that QTLs for growth are often age-dependent, just as reported in the present study. Thus, contrary to claims in the present manuscript (e.g., lines 35-36, 130-142, 584) the fact that the genetic architecture of growth is ontogenetically heterogeneous is empirically well-established and not novel, not even in fish.

To my mind, this does not diminish the value of the present study. This manuscript provides a nice illustration of this general principle in sticklebacks. But the authors need to acknowledge the large literature that already exists on this topic, and they need to put their results into the context of this existing knowledge in other species rather than claiming novelty.

Here are just a few of many more studies on age-dependent QTLs in farm animals:

Carlborg, O., Kerje, S., Schütz, K., Jacobsson, L., Jensen, P., & Andersson, L. (2003). A global search reveals epistatic interaction between QTL for early growth in the chicken. *Genome research*, 13(3), 413–421. <https://doi.org/10.1101/gr.528003>

Hadjipavlou, G. and Bishop, S.C. (2009), Age-dependent quantitative trait loci affecting growth traits in Scottish Blackface sheep. *Animal Genetics*, 40: 165-175. <https://doi.org/10.1111/j.1365-2052.2008.01814.x>

Podisi, B.K., Knott, S.A., Burt, D.W. et al. Comparative analysis of quantitative trait loci for body weight, growth rate and growth curve parameters from 3 to 72 weeks of age in female chickens of a broiler–layer cross. *BMC Genet* 14, 22 (2013). <https://doi.org/10.1186/1471-2156-14-22>

Miyako Kodama, Jeffrey J. Hard, Kerry A. Naish. 2018. Mapping of quantitative trait loci for temporal growth and age at maturity in coho salmon: Evidence for genotype-by-sex interactions. *Marine Genomics* (38): 33-44. <https://doi.org/10.1016/j.margen.2017.07.004>.

We thank the Referee for these helpful references. We acknowledge that important studies in the field of domestic species' genetics were left out of the citation list in the previous version of the manuscript. It was not our intention to brush away these studies and claim novelty for the results but rather to focus more on the quantitative genetics literature from the wild.

As mentioned in our response above, we have now toned down any claims for novelty of our findings (i.e., age-specific genetic architecture) but instead, emphasized that this well-known aspect has not been put in the context of adaptive (wild) evolution to the best of our knowledge. We have also added text in our Discussion (L348-352 in the track-changes version of the revision) to put our results in a broader context and particularly, to show that our results are in line with many previous studies showing heterogeneous genetic bases for early and late growth traits.

2) Perhaps I made a mistake somewhere but the files I could download did not include the figures and tables. Hence, I cannot comment on the figures and tables and the result presented therein.

We are sorry to read that the Referee could not access the figures and tables. After checking our submission and with the editorial board we confirm that they were well attached. We are attaching them again to this revision and hope the issue will be solved.

Detailed Points:

Before addressing these points, we would like to point out that we were not able to match the Referee's line numbers to our main text. We believe that something wrong must have happened during the submission process which somehow led to the Referee not being able to access our figures/tables and perhaps also to our original line numbering.

We thus tried our best to address these comments by deducing the line numbers to which they referred to and apologize if they were not the right ones. We added to our responses the line numbers at which our changes were made (line numbers correspond to the track-changes version of our revised ms).

- Abstract: The Abstract would profit from more substantial changes in light of the other changes in the manuscript.

Thank you, we have now reworded the abstract.

- L. 323: 'Gaussian' should perhaps be capitalised?

This has been corrected (L169)

- L. 462: Should read 'grandparental'.

Sorry we could not find where the change should be implemented.

- L. 578: I wasn't sure how to reconcile the statement here, that phenotypic growth trajectories are very similar, with Figure S1 and with the significant additive genetic variation found in this study. The latter two suggest to me that there was substantial variation in growth trajectories, which goes nicely with the QTL results. Thus, to my mind and contrary to the wording here, there is no fundamental contradiction between genetic and phenotypic patterns.

We agree with the Referee here and realize that our wording was misleading: we indeed think too that there is substantial variation in growth trajectories and that this result concurs with the QTL results. We meant here – in continuation of the opening sentence of our Discussion – that despite the similar phenotypic end point reached in the different crosses (i.e., relatively large size due to the effects of pond alleles) we observe heterogeneity in the genetic architecture. We reworded this sentence to avoid confusion which now reads: “*Our results demonstrate that the QTL contributing to size were different across ages both within and among the three crosses studied here.*” (L306)